# Neural Collapse Inspired Feature-Classifier Alignment for Few-Shot Class Incremental Learning

**Yibo Yang**[1*], **Haobo Yuan**[2*], **Xiangtai Li**[3], **Zhouchen Lin**[3,4,5†], **Philip Torr**[6], **Dacheng Tao**[1]

[1]JD Explore Academy    [2]School of Computer Science, Wuhan University
[3]National Key Lab of General AI, School of Intelligence Science and Technology, Peking University
[4]Institute for Artificial Intelligence, Peking University    [5]Peng Cheng Laboratory    [6]University of Oxford
†: corresponding author    *: equal contribution

## Abstract

Few-shot class-incremental learning (FSCIL) has been a challenging problem as only a few training samples are accessible for each novel class in the new sessions. Finetuning the backbone or adjusting the classifier prototypes trained in the prior sessions would inevitably cause a misalignment between the feature and classifier of old classes, which explains the well-known catastrophic forgetting problem. In this paper, we deal with this misalignment dilemma in FSCIL inspired by the recently discovered phenomenon named *neural collapse*, which reveals that the last-layer features of the same class will collapse into a vertex, and the vertices of all classes are aligned with the classifier prototypes, which are formed as a simplex equiangular tight frame (ETF). It corresponds to an optimal geometric structure for classification due to the maximized Fisher Discriminant Ratio. We propose a neural collapse inspired framework for FSCIL. A group of classifier prototypes are pre-assigned as a simplex ETF for the whole label space, including the base session and all the incremental sessions. During training, the classifier prototypes are not learnable, and we adopt a novel loss function that drives the features into their corresponding prototypes. Theoretical analysis shows that our method holds the neural collapse optimality and does not break the feature-classifier alignment in an incremental fashion. Experiments on the miniImageNet, CUB-200, and CIFAR-100 datasets demonstrate that our proposed framework outperforms the state-of-the-art performances. Code address: `https://github.com/NeuralCollapseApplications/FSCIL`

## 1 Introduction

Learning incrementally and learning with few-shot data are common in the real-world implementations, and in many applications, such as robotics, the two demands emerge simultaneously. Despite the great success in a closed label space, it is still challenging for a deep learning model to learn new classes continually with only limited samples (LeCun et al., 2015). To this end, few-shot class-incremental learning (FSCIL) was proposed to tackle this problem (Tao et al., 2020b).

Compared with few-shot learning (Ravi & Larochelle, 2017; Vinyals et al., 2016), FSCIL transfers a trained model into new label spaces incrementally. It also differs from incremental learning (Cauwenberghs & Poggio, 2000; Li & Hoiem, 2017; Rebuffi et al., 2017) in that there are only a few (usually 5) samples accessible for each new class in the incremental sessions. For each session's evaluation, the model is required to infer test images coming from all the classes that have been encountered. The base session of FSCIL contains a large label space and sufficient training samples, while each incremental session only has a few novel classes and labeled images. It poses the notorious *catastrophic forgetting* problem (Goodfellow et al., 2013) because the novel sessions have no access to the data of the previous sessions.

Due to the importance and difficulty, FSCIL has attracted much research attention. The initial solutions to FSCIL finetune the network on new session data with distillation schemes to reduce the forgetting of old classes (Tao et al., 2020b; Dong et al., 2021). However, the few-shot data in novel sessions can easily induce over-fitting. Following studies favor training a backbone network on the

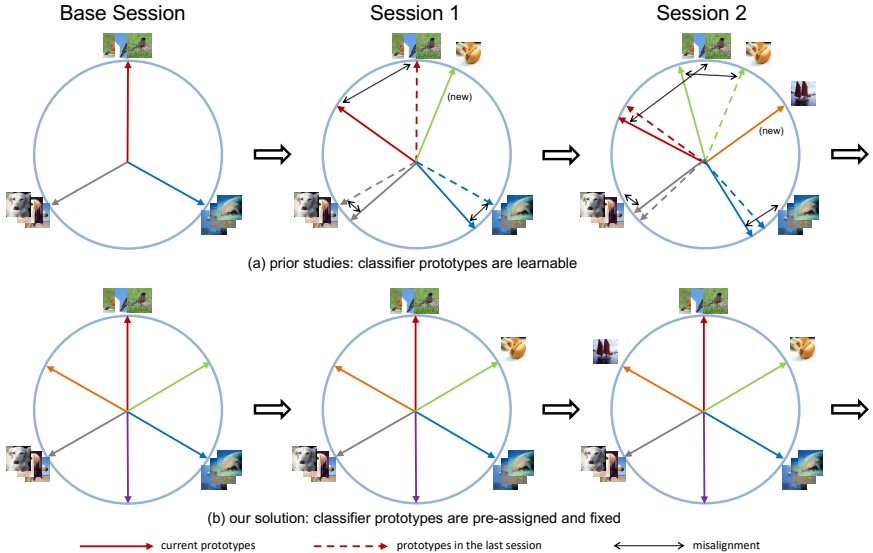

Figure 1: A popular choice in prior studies is to evolve the old-class prototypes via delicate design of loss or regularizer to keep them separated from novel-class prototypes, but will cause misalignment. As a comparison, we pre-assign and fix an optimal feature-classifier alignment, and then train a model towards the same neural collapse optimality in each session to avoid target conflict.

base session as a feature extractor (Zhang et al., 2021; Hersche et al., 2022; Akyürek et al., 2022). For novel sessions, the backbone network is fixed and a group of novel-class prototypes (classifier vectors) are learned incrementally. But as shown in Figure 1 (a), the newly added prototypes may lie close to the old-class prototypes, which impedes the ability to discriminate between the old-class and the novel-class samples in evaluation. As a result, adjusting the classifier prototypes is always necessary for two goals: (*i*) keep a sufficient distance between the old-class and the novel-class prototypes; (*ii*) prevent the adjusted old-class prototypes from shifting far away from their original positions. However, the two goals rely on sophisticated loss functions or regularizers (Chen & Lee, 2021; Hersche et al., 2022; Akyürek et al., 2022), and are hard to attain simultaneously without qualification. Besides, as shown in Figure 1 (a), there will be a misalignment between the adjusted classifier and the fixed features of old classes. A recent study proposes to reserve feature space for novel classes to circumvent their conflict with old classes (Zhou et al., 2022a), but an optimal feature-classifier alignment is hard to be guaranteed with learnable classifier (Pernici et al., 2021).

We point out that it is the misalignment dilemma between feature and classifier that causes the catastrophic forgetting problem of old classes. If a backbone network is finetuned in novel sessions, the features of old classes will be easily deviated from their classifier prototypes. Alternatively, when a backbone network is fixed and a group of new prototypes for novel classes are learned incrementally, the adjustment of old-class prototypes will also induce misalignments with their fixed features. In this paper, we pose and study the following question,

*"Can we look for and pre-assign an optimal feature-classifier alignment such that the model is optimized towards the fixed optimality, so avoids conflict among sessions?"*

### 1.1 MOTIVATIONS AND CONTRIBUTIONS

Neural collapse is a recently discovered phenomenon that at the terminal phase of training (after 0 training error rate), the last-layer features of the same class will collapse into a single vertex, and the vertices of all classes will be aligned with their classifier prototypes and be formed as a simplex equiangular tight frame (ETF) (Papyan et al., 2020). A simplex ETF is a geometric structure of $K$ vectors in $\mathbb{R}^d$, $d \geq K-1$. All vectors have the same $\ell_2$ norm of 1 and any pair of two different vectors has an inner product of $-\frac{1}{K-1}$, which corresponds to the largest possible angle of $K$ equiangular vectors. Particularly when $d = K-1$, a simplex ETF reduces to a regular simplex such as triangle and tetrahedron. It describes an optimal geometric structure for classification due to the minimized within-class variance and the maximized between-class variance (Martinez & Kak, 2001), which indicates that the Fisher Discriminant Ratio (Fisher, 1936; Rao, 1948) is maximized. Following studies aim to theoretically explain this phenomenon (Fang et al., 2021; Han et al., 2022).

It is expected that imperfect training condition, such as imbalance, cannot induce neural collapse and will cause deteriorated performance (Fang et al., 2021; Yang et al., 2022b). Training in an incremental fashion will also break the neural collapse optimality. Since neural collapse offers us an optimal structure where features and their classifier prototypes are aligned, we can pre-assign such a structure and learn the model towards the optimality. Inspired by this insight, in this paper, we initialize a group of classifier prototypes $\hat{\mathbf{W}}_{\text{ETF}} \in \mathbb{R}^{d \times (K_0 + K')}$ as a simplex ETF for the whole label space, where $K_0$ is the number of classes in the base session and $K'$ is the number of classes in all the incremental sessions. As shown in Figure 1 (b), it serves as the optimization target and keeps fixed throughout all sessions training. We append a projection layer after the backbone network and store the mean latent feature of each class output by the backbone in a memory. In the training of incremental sessions, we only finetune the projection layer using a novel loss function that drives the final features towards their corresponding target prototypes. Without bells and whistles, our method achieves superior performances and relieves the catastrophic forgetting problem.

The contributions of this paper can be summarized as follows:

- To relieve the misalignment dilemma in FSCIL, we propose to pre-assign an optimal alignment inspired by neural collapse as a fixed target throughout the incremental learning. Our model is trained towards the same optimality to avoid optimization conflict among sessions.

- We fix the prototypes and apply a novel loss function that only finetunes a projection layer to drive the output features into their corresponding prototypes. Theoretical and empirical analyses show that our method better holds the neural collapse optimality.

- Experiments on miniImageNet, CIFAR-100, and CUB-200 demonstrate that our method is able to surpass the state-of-the-art performances. In particular, our method achieves an average accuracy improvement of more than 3.5% over a recent strong baseline on both miniImageNet and CIFAR-100.

## 2   RELATED WORK

**Few-shot class-incremental learning (FSCIL).** As a variant of class-incremental learning (CIL) (Cauwenberghs & Poggio, 2000; Li & Hoiem, 2017; Rebuffi et al., 2017), FSCIL only has a few novel-classes and training data in each incremental session (Tao et al., 2020b; Dong et al., 2021), which increases the tendency of overfitting on novel classes (Snell et al., 2017; Sung et al., 2018). Both CIL and FSCIL require a delicate balance between well adapting a model to novel classes and less forgetting of old classes (Zhao et al., 2021). A popular choice is to use meta learning (Yoon et al., 2020; Chi et al., 2022; Zhou et al., 2022b). Some studies try to make base and incremental sessions compatible via pseudo-feature (Cheraghian et al., 2021b; Zhou et al., 2022a), augmentation (Peng et al., 2022), or looking for a flat minima (Shi et al., 2021). For training in incremental sessions, the new prototypes for novel classes should be separable from the old-class prototypes. Meanwhile, the adjustment of old-class prototypes should not induce large shifts. Current studies widely rely on evolving the prototypes (Zhang et al., 2021; Zhu et al., 2021a) or sophisticated designs of loss and regularizer (Ren et al., 2019; Hou et al., 2019; Tao et al., 2020a; Joseph et al., 2022; Lu et al., 2022; Hersche et al., 2022; Chen & Lee, 2021; Akyürek et al., 2022; Yang et al., 2022a). However, the two goals have inherent conflict, and a tough effort to balance the loss terms is necessary. In contrast, our method pre-assigns and fixes a feature-classifier alignment as an optimality. A model is trained towards the same target in all sessions. **We only use a single loss without any regularizer**.

**Neural collapse.** Neural collapse describes an elegant geometric structure of the last-layer feature and classifier in a well-trained model (Papyan et al., 2020). It inspires later studies to theoretically explain this phenomenon. Based on a simplified model that only considers the last-layer optimization, neural collapse is proved to be the global optimality of balanced training with the CE (Weinan & Wojtowytsch, 2020; Graf et al., 2021; Lu & Steinerberger, 2020; Fang et al., 2021; Zhu et al., 2021b; Ji et al., 2022) and the MSE (Mixon et al., 2020; Poggio & Liao, 2020; Zhou et al., 2022c; Han et al., 2022; Tirer & Bruna, 2022) loss functions. Recent studies try to induce neural collapse in imbalanced training by fixing a classifier (Yang et al., 2022b; Zhong et al., 2023) or novel loss (Xie et al., 2023). Our method is inspired by Yang et al. (2022b), but we apply the classifier in an incremental fashion. Galanti et al. (2022) show that neural collapse is still valid when transferring a model into new samples or classes. To the best of our knowledge, **we are the first to study FSCIL from the neural collapse perspective, which offers our method sound interpretability**.

## 3 BACKGROUND

### 3.1 FEW-SHOT CLASS-INCREMENTAL LEARNING (FSCIL)

In real-world applications, one often needs to adapt a model to data coming from a new label space with only a few labeled samples. FSCIL trains a model incrementally on a sequence of training datasets $\{\mathcal{D}^{(0)}, \mathcal{D}^{(1)}, \ldots, \mathcal{D}^{(T)}\}$, where $\mathcal{D}^{(t)} = \{(\boldsymbol{x}_i, y_i)\}_{i=1}^{|\mathcal{D}^{(t)}|}$, $\mathcal{D}^{(0)}$ is the base session, and $T$ the number of incremental sessions. The base session $\mathcal{D}^{(0)}$ usually contains a large label space $\mathcal{C}^{(0)}$ and sufficient training images for each class $c \in \mathcal{C}^{(0)}$. In each incremental session $\mathcal{D}^{(t)}$, $t > 0$, there are only a few labeled images and we have $|\mathcal{D}^{(t)}| = pq$, where $p$ is the number of classes and $q$ is the number samples per novel class, known as $p$-way $q$-shot. The label space $\mathcal{C}^{(t)}$ has no overlap with any other session, *i.e.,* $\mathcal{C}^{(t)} \cap \mathcal{C}^{(t')} = \emptyset$, $\forall t' \neq t$. For any incremental session $t > 0$, we only have access to the data in $\mathcal{D}^{(t)}$, and the training sets of the previous sessions are not available. For evaluation in session $t$, the test dataset comes from all the encountered classes in the previous and current sessions [1], *i.e.* the label space of $\cup_{i=0}^{t} \mathcal{C}^{(i)}$.

Therefore, FSCIL suffers from severe data scarcity and imbalance. It requires a model to be adaptable to novel classes, and meanwhile keep the ability on old classes.

### 3.2 NEURAL COLLAPSE

Neural collapse refers to a phenomenon at the terminal phase of training (after 0 training error rate) on balanced data (Papyan et al., 2020). It reveals a geometric structure formed by the last-layer feature and classifier that can be defined as:

**Definition 1 (Simplex Equiangular Tight Frame)** *A simplex Equiangular Tight Frame (ETF) refers to a matrix that is composed of $K$ vectors in $\mathbb{R}^d$ and satisfies:*

$$\mathbf{E} = \sqrt{\frac{K}{K-1}} \mathbf{U} \left( \mathbf{I}_K - \frac{1}{K} \mathbf{1}_K \mathbf{1}_K^T \right), \tag{1}$$

*where $\mathbf{E} = [\mathbf{e}_1, \cdots, \mathbf{e}_K] \in \mathbb{R}^{d \times K}$, $\mathbf{U} \in \mathbb{R}^{d \times K}$ allows a rotation and satisfies $\mathbf{U}^T \mathbf{U} = \mathbf{I}_K$, $\mathbf{I}_K$ is the identity matrix, and $\mathbf{1}_K$ is an all-ones vector.*

*All column vectors in $\mathbf{E}$ have the same $\ell_2$ norm and any pair has an inner produce of $-\frac{1}{K-1}$,* i.e.,

$$\mathbf{e}_{k_1}^T \mathbf{e}_{k_2} = \frac{K}{K-1} \delta_{k_1, k_2} - \frac{1}{K-1}, \quad \forall k_1, k_2 \in [1, K], \tag{2}$$

*where $\delta_{k_1, k_2} = 1$ when $k_1 = k_2$, and 0 otherwise.*

The neural collapse phenomenon includes the following four properties:

**(NC1)**: The last-layer features of the same class will collapse into their within-class mean, *i.e.,* the covariance $\Sigma_W^{(k)} \to \mathbf{0}$, where $\Sigma_W^{(k)} = \text{Avg}_i \{ (\boldsymbol{\mu}_{k,i} - \boldsymbol{\mu}_k)(\boldsymbol{\mu}_{k,i} - \boldsymbol{\mu}_k)^T \}$, $\boldsymbol{\mu}_{k,i}$ is the feature of sample $i$ in class $k$, and $\boldsymbol{\mu}_k$ is the within-class mean of class $k$ features;

**(NC2)**: The within-class means of all classes centered by the global mean will converge to the vertices of a simplex ETF defined in Definition 1, *i.e.,* $\hat{\boldsymbol{\mu}}_k$, $1 \leq k \leq K$ satisfy Eq. (2), where $\hat{\boldsymbol{\mu}}_k = (\boldsymbol{\mu}_k - \boldsymbol{\mu}_G)/\|\boldsymbol{\mu}_k - \boldsymbol{\mu}_G\|$ and $\boldsymbol{\mu}_G$ is the global mean;

**(NC3)**: The within-class means centered by the global mean will be aligned with (parallel to) their corresponding classifier weights, which means the classifier weights will converge to the same simplex ETF, *i.e.,* $\hat{\boldsymbol{\mu}}_k = \boldsymbol{w}_k/\|\boldsymbol{w}_k\|$, $1 \leq k \leq K$, where $\boldsymbol{w}_k$ is the classifier weight of class $k$;

**(NC4)**: When **(NC1)**-**(NC3)** hold, the model prediction using logits can be simplified to the nearest class centers [2], *i.e.,* $\arg\max_k \langle \boldsymbol{\mu}, \boldsymbol{w}_k \rangle = \arg\min_k \|\boldsymbol{\mu} - \boldsymbol{\mu}_k\|$, where $\langle \cdot \rangle$ is the inner product operator, $\boldsymbol{\mu}$ is the last-layer feature of a sample for prediction.

Neural collapse corresponds to an optimal feature-classifier alignment for classification due to the maximized Fisher Discriminant Ratio (between-class variance to within-class variance).

---

[1] Different from task-incremental learning, we do not know which session a test sample comes from.

[2] We omit the bias term in a linear classifier layer for simplicity.

# 4 METHOD

Neural collapse tells us an optimal geometric structure for classification problems where the last-layer features and classifier prototype of the same class are aligned, and those of different classes are **maximally separated**. However, this structure will be broken in imperfect training conditions, such as imbalanced training data (Fang et al., 2021; Yang et al., 2022b). As illustrated in Figure 1 (a), training in an incremental fashion will also break the neural collapse optimality. Inspired by this perspective, what we should do for FSCIL is to keep the neural collapse inspired feature-classifier alignment as sound as possible. Concretely, we adopt a fixed classifier and a novel loss function as described in Section 4.1 and Section 4.2, respectively. We introduce our framework for FSCIL in Section 4.3. Finally, in Section 4.4, we conduct theoretical analysis to show how our method better holds the neural collapse optimality in an incremental fashion.

## 4.1 ETF CLASSIFIER

Assume that the base session contains a label space of $K_0$ classes, each incremental session has $p$ classes, and we have $T$ incremental sessions in total. The whole label space of this FSCIL problem has $K_0 + K'$ classes, where $K' = Tp$, *i.e.,* we need to learn a model that can recognize samples from $K_0 + K'$ classes. We denote a backbone network as $f$, and then we have $\boldsymbol{\mu} = f(\boldsymbol{x}, \theta_f)$, where $\boldsymbol{\mu} \in \mathbb{R}^d$ is the output feature of input $\boldsymbol{x}$, and $\theta_f$ is the backbone network parameters.

A popular choice in current studies learns $f$ and $\mathbf{W}^{(0)}$ using the base session data, where $\mathbf{W}^{(0)} \in \mathbb{R}^{d \times K_0}$ is the classifier prototypes for base classes. In incremental sessions $t > 0$, $f$ is fixed as a feature extractor and only $\mathbf{W}^{(t)} \in \mathbb{R}^{d \times p}$ for novel classes is learnable. As shown in Figure 1 (a), one need to adjust $\{\mathbf{W}^{(0)}, \cdots, \mathbf{W}^{(t)}\}$ via sophisticated loss or regularizer to ensure separation among these prototypes (Akyürek et al., 2022; Hersche et al., 2022). But it will inevitably introduce mis-alignment between the adjusted prototypes and the fixed features of old classes. It is an underlying reason for the catastrophic forgetting problem (Joseph et al., 2022).

Since neural collapse describes an optimal geometric structure of the last-layer feature and classifier, we pre-assign such an optimality by fixing a learnable classifier as the structure instructed by neural collapse. Following Yang et al. (2022b), we adopt an ETF classifier that initializes a classifier as a simplex ETF and fixes it during training. The difference lies in that we use it in an incremental fashion. Concretely, we randomly initialize classifier prototypes $\hat{\mathbf{W}}_{\text{ETF}} \in \mathbb{R}^{d \times (K_0 + K')}$ by Eq. (1) for the whole label space, *i.e.,* the union of classes in all session, $\cup_{i=0}^{T} \mathcal{C}^{(i)}$. We have $K_0 = |\mathcal{C}^{(0)}|$ and $K' = \sum_{i=1}^{T} |\mathcal{C}^{(i)}| = Tp$. Then any pair $(k_1, k_2)$ of classifier prototypes in $\hat{\mathbf{W}}_{\text{ETF}}$ satisfies:

$$\hat{\mathbf{w}}_{k_1}^T \hat{\mathbf{w}}_{k_2} = \frac{K_0 + K'}{K_0 + K' - 1} \delta_{k_1, k_2} - \frac{1}{K_0 + K' - 1}, \quad \forall k_1, k_2 \in [1, K_0 + K'], \tag{3}$$

where $\hat{\mathbf{w}}_{k_1}$ and $\hat{\mathbf{w}}_{k_2}$ are two column vectors in $\hat{\mathbf{W}}_{\text{ETF}}$. Our ETF classifier ensures that the prototypes of the whole label space have the maximal pair-wise separation. It serves as a fixed target along the incremental training to avoid conflict among sessions. We only need to learn a model whose output features are aligned with this pre-assigned structure.

## 4.2 DOT-REGRESSION LOSS

The gradient of cross entropy (CE) loss with respect to the last-layer feature is composed of a `pull` term that drives the feature into its classifier prototype of the same class, and a `push` term that pushes it away from the prototypes of different classes. As pointed out by Yang et al. (2022b), when the classifier prototypes are fixed as an optimality, the `pull` term is always accurate towards the solution, and we can drop the `push` gradient that may be inaccurate. Accordingly, we adopt a novel loss named dot-regression (DR) loss that can be formulated as (Yang et al., 2022b):

$$\mathcal{L}\left(\hat{\boldsymbol{\mu}}_i, \hat{\mathbf{W}}_{\text{ETF}}\right) = \frac{1}{2} \left(\hat{\mathbf{w}}_{y_i}^T \hat{\boldsymbol{\mu}}_i - 1\right)^2, \tag{4}$$

where $\hat{\boldsymbol{\mu}}_i$ is the normalized feature, *i.e.,* $\hat{\boldsymbol{\mu}}_i = \boldsymbol{\mu}_i / \|\boldsymbol{\mu}_i\|$, $\boldsymbol{\mu}_i = f(\boldsymbol{x}_i, \theta_f)$, $y_i$ is the label of input $\boldsymbol{x}_i$, $\hat{\mathbf{w}}_{y_i}$ is the fixed prototype in $\hat{\mathbf{W}}_{\text{ETF}}$ for class $y_i$, and we have $\|\hat{\mathbf{w}}_{y_i}\| = 1$ by Eq. (3). The total loss is an average over a batch of input $\boldsymbol{x}_i$. The gradient of Eq. (4) with respect to $\hat{\boldsymbol{\mu}}_i$ takes the form of:

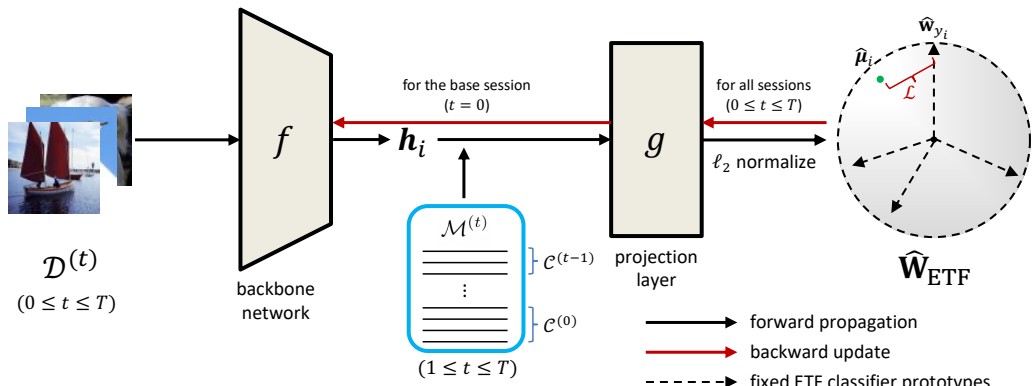

Figure 2: An illustration of our NC-FSCIL. $h_i$ is the intermediate feature from the backbone network $f$. $\hat{\mu}_i$ is the normalized output feature after the projection layer $g$. $\hat{\mathbf{W}}_{\mathrm{ETF}}$ is the ETF classifier that contains prototypes of the whole label space and serves as a fixed target throughout the incremental training. $\mathcal{L}$ denotes the dot-regression loss function. $f$ is frozen in the incremental sessions ($1 \leq t \leq T$). A small memory of old-class features is widely adopted in prior studies such as Cheraghian et al. (2021a), Chen & Lee (2021), Akyürek et al. (2022), and Hersche et al. (2022).

$\partial \mathcal{L} / \partial \hat{\mu}_i = -(1 - \cos \angle (\hat{\mu}_i, \hat{\mathbf{w}}_{y_i})) \hat{\mathbf{w}}_{y_i}$. It is shown that the gradient pulls feature $\hat{\mu}_i$ towards the direction of $\hat{\mathbf{w}}_{y_i}$, which is a pre-assigned target prototype. Finally, the converged features will be aligned with $\hat{\mathbf{W}}_{\mathrm{ETF}}$, and thus the geometric structure instructed by neural collapse is attained. The theoretical advantage of the DR loss has been proved in Yang et al. (2022b). In experiments, we will compare the DR loss with the CE loss to show its effectiveness in FSCIL.

## 4.3 NC-FSCIL

Based on the ETF classifier and the DR loss, we now introduce our neural collapse inspired framework for few-shot class-incremental learning (NC-FSCIL). As shown in Figure 2, our model is composed of two components, a backbone network $f$ and a projection layer $g$. The backbone network $f$ takes the training data $x_i$ as input, and outputs an intermediate feature $h_i$. The projection layer $g$ can be a linear transformation or an MLP block following Hersche et al. (2022); Peng et al. (2022). It projects the intermediate feature $h_i$ into $\mu_i$. Finally, we perform an $\ell_2$ normalization on $\mu_i$ to get the output feature $\hat{\mu}_i$, *i.e.*,

$$\hat{\mu}_i = \mu_i / \|\mu_i\|, \quad \mu_i = g(h_i, \theta_g), \quad h_i = f(x_i, \theta_f), \tag{5}$$

where $\theta_f$ and $\theta_g$ denote the parameters of the backbone network and the projection layer, respectively. We use the normalized output feature $\hat{\mu}_i$ to compute error signal by Eq. (4).

In the base session $t = 0$, we jointly train both $f$ and $g$ using the base session data. The empirical risk to minimize in the base session can be formulated as:

$$\min_{\theta_f, \theta_g} \quad \frac{1}{|\mathcal{D}^{(0)}|} \sum_{(x_i, y_i) \in \mathcal{D}^{(0)}} \mathcal{L}\left(\hat{\mu}_i, \hat{\mathbf{W}}_{\mathrm{ETF}}\right), \tag{6}$$

where $\hat{\mathbf{W}}_{\mathrm{ETF}}$ is the pre-assigned ETF classifier as introduced in Section 4.1, $\mathcal{L}$ is the DR loss as introduced in Section 4.2, and $\hat{\mu}_i$ is a function of $f$ and $g$ as shown in Eq. (5).

In each incremental session $1 \leq t \leq T$, we fix the backbone network $f$ as a feature extractor, and only finetune the projection layer $g$. As a widely adopted practice in FSCIL studies, a small memory of samples or features of old classes can be retained to relieve the overfitting on novel classes (Cheraghian et al., 2021a; Chen & Lee, 2021; Akyürek et al., 2022; Hersche et al., 2022). Following Hersche et al. (2022), we only keep a memory $\mathcal{M}^{(t)}$ of the mean intermediate feature $h_c$ for each old class $c$. Concretely, we have,

$$\mathcal{M}^{(t)} = \{h_c | c \in \cup_{j=0}^{t-1} \mathcal{C}^{(j)}\}, \quad h_c = \mathrm{Avg}_i\{f(x_i, \theta_f) | y_i = c\}, \quad 1 \leq t \leq T, \tag{7}$$

where $f$ has been fixed after the base session. Then we use $\mathcal{D}^{(t)}$ as the input of $f$, and $\mathcal{M}^{(t)}$ as the input of $g$ to finetune the projection layer $g$. The empirical risk to minimize in incremental sessions

can be formulated as:

$$\min_{\theta_g} \quad \frac{1}{|\mathcal{D}^{(t)}| + |\mathcal{M}^{(t)}|} \left( \sum_{(\boldsymbol{x}_i, y_i) \in \mathcal{D}^{(t)}} \mathcal{L}\left(\hat{\boldsymbol{\mu}}_i, \hat{\mathbf{W}}_{\text{ETF}}\right) + \sum_{(\boldsymbol{h}_c, y_c) \in \mathcal{M}^{(t)}} \mathcal{L}\left(\hat{\boldsymbol{\mu}}_c, \hat{\mathbf{W}}_{\text{ETF}}\right) \right), \quad (8)$$

where $\hat{\boldsymbol{\mu}}_i$, and $\hat{\boldsymbol{\mu}}_c$ are the output features of $\boldsymbol{x}_i$ and $\boldsymbol{h}_c$, respectively, $|\mathcal{D}^{(t)}|$ is the number of training samples in session $t$, and we have $|\mathcal{M}^{(t)}| = \sum_{j=0}^{t-1} |\mathcal{C}^{(j)}|$. **Thanks to our pre-assigned alignment, we do not rely on any regularizer in our training**.

In the evaluation of session $t$, we predict an input $\boldsymbol{x}$ based on the inner product between its output feature $\hat{\boldsymbol{\mu}}$ and the ETF classifier prototypes: $\arg\max_k \langle \hat{\boldsymbol{\mu}}, \hat{\mathbf{w}}_k \rangle$, $\forall 1 \leq k \leq K_0 + K'$.

## 4.4 THEORETICAL SUPPORTS

We perform our theoretical work based on a simplified model that drops the backbone network and only keeps the last-layer features and classifier prototypes as independent optimization variables. This simplification has been widely adopted in prior studies to facilitate analysis (Graf et al., 2021; Fang et al., 2021; Zhu et al., 2021b). We investigate the neural collapse optimality of an incremental problem of $T$ sessions with our ETF classifier. Concretely, we consider the following problem,

$$\min_{\mathbf{M}^{(t)}} \quad \frac{1}{N^{(t)}} \sum_{k=1}^{K^{(t)}} \sum_{i=1}^{n_k} \mathcal{L}\left(\mathbf{m}_{k,i}^{(t)}, \hat{\mathbf{W}}_{\text{ETF}}\right), \ 0 \leq t \leq T, \quad (9)$$

$$s.t. \quad \|\mathbf{m}_{k,i}^{(t)}\|^2 \leq 1, \quad \forall 1 \leq k \leq K^{(t)}, \ 1 \leq i \leq n_k,$$

where $\mathbf{m}_{k,i}^{(t)} \in \mathbb{R}^d$ denotes a feature variable that belongs to the $i$-th sample of class $k$ in session $t$, $n_k$ is number of samples in class $k$, $K^{(t)}$ is number of classes in session $t$, $N^{(t)}$ is the number of samples in session $t$, *i.e.*, $N^{(t)} = \sum_{k=1}^{K^{(t)}} n_k$, and $\mathbf{M}^{(t)} \in \mathbb{R}^{d \times N^{(t)}}$ denotes a collection of $\boldsymbol{m}_{k,i}^{(t)}$. $\hat{\mathbf{W}}_{\text{ETF}} \in \mathbb{R}^{d \times K}$ refers to the ETF classifier for the whole label space as introduced in Section 4.1, and we have $K = \sum_{t=0}^{T} K^{(t)}$. $\mathcal{L}$ can be both the cross entropy and the dot regression loss functions.

**Theorem 1** *Let $\hat{\mathbf{M}}^{(t)}$ denote the global minimizer of Eq. (9) by optimizing the model incrementally from $t = 0$, and we have $\hat{\mathbf{M}} = [\hat{\mathbf{M}}^{(0)}, \cdots, \hat{\mathbf{M}}^{(T)}] \in \mathbb{R}^{d \times \sum_{t=0}^{T} N^{(t)}}$. When $\mathcal{L}$ in Eq. (9) is CE or DR loss, for any column vector $\hat{\mathbf{m}}_{k,i}$ in $\hat{\mathbf{M}}$ whose class label is $k$, we have:*

$$\|\hat{\mathbf{m}}_{k,i}\| = 1, \ \hat{\mathbf{m}}_{k,i}^T \hat{\mathbf{w}}_{k'} = \frac{K}{K-1}\delta_{k,k'} - \frac{1}{K-1}, \ \forall k, k' \in [1, K], \ 1 \leq i \leq n_k, \quad (10)$$

*where $K = \sum_{t=0}^{T} K^{(t)}$ denotes the total number of classes of the whole label space, $\delta_{k,k'} = 1$ when $k = k'$ and 0 otherwise, and $\hat{\mathbf{w}}_{k'}$ is the prototype of class $k'$ in $\hat{\mathbf{W}}_{\text{ETF}}$.*

The proof of Theorem 1 can be found in Appendix A. Eq. (10) indicates that the global minimizer $\hat{\mathbf{M}}$ of Eq. (9) satisfies the neural collapse condition, *i.e.*, features of the same class collapse into a single vertex, and the vertices of all classes are aligned with $\hat{\mathbf{W}}_{\text{ETF}}$ as a simplex ETF. It is shown that the feature space is equally separated by prototypes of all classes. More importantly, in problem Eq. (9), the number of classes $K^{(t)}$ among $T + 1$ sessions and the number of samples $n_k$ among $K$ classes can be imbalanced, which corresponds to the challenging demand of FSCIL.

## 5 EXPERIMENTS

In this section, we test our method on FSCIL benchmark datasets including miniImageNet (Russakovsky et al., 2015), CIFAR-100 (Krizhevsky et al., 2009), and CUB-200 (Wah et al., 2011), and compare it with state-of-the-art methods. We also perform ablation studies to validate the effects of ETF classifier and DR loss. Finally, we show the feature-classifier structure achieved by our method.

## 5.1 IMPLEMENTATION DETAILS

Please refer to Appendix B for our implementation details.

Table 1: Performance of FSCIL in each session on miniImageNet and comparison with other studies. The top rows list class-incremental learning and few-shot learning results implemented by Tao et al. (2020b); Zhang et al. (2021) in the FSCIL setting. "Average Acc." is the average accuracy of all sessions. "Final Improv." calculates the improvement of our method in the last session. * indicates that the method saves the within-class feature mean of each class for training or inference.

| Methods | Accuracy in each session (%) ↑ | | | | | | | | | Average | Final |
| | 0 | 1 | 2 | 3 | 4 | 5 | 6 | 7 | 8 | Acc. | Improv. |
|---|---|---|---|---|---|---|---|---|---|---|---|
| iCaRL (Rebuffi et al., 2017) | 61.31 | 46.32 | 42.94 | 37.63 | 30.49 | 24.00 | 20.89 | 18.80 | 17.21 | 33.29 | **+41.1** |
| NCM (Hou et al., 2019) | 61.31 | 47.80 | 39.30 | 31.90 | 25.70 | 21.40 | 18.70 | 17.20 | 14.17 | 30.83 | **+44.14** |
| D-Cosine (Vinyals et al., 2016) | 70.37 | 65.45 | 61.41 | 58.00 | 54.81 | 51.89 | 49.10 | 47.27 | 45.63 | 55.99 | **+12.68** |
| *TOPIC (Tao et al., 2020b) | 61.31 | 50.09 | 45.17 | 41.16 | 37.48 | 35.52 | 32.19 | 29.46 | 24.42 | 39.64 | **+33.89** |
| *IDLVQ (Chen & Lee, 2021) | 64.77 | 59.87 | 55.93 | 52.62 | 49.88 | 47.55 | 44.83 | 43.14 | 41.84 | 51.16 | **+16.47** |
| Self-promoted (Zhu et al., 2021a) | 61.45 | 63.80 | 59.53 | 55.53 | 52.50 | 49.60 | 46.69 | 43.79 | 41.92 | 52.76 | **+16.39** |
| CEC (Zhang et al., 2021) | 72.00 | 66.83 | 62.97 | 59.43 | 56.70 | 53.73 | 51.19 | 49.24 | 47.63 | 57.75 | **+10.68** |
| *LIMIT (Zhou et al., 2022b) | 72.32 | 68.47 | 64.30 | 60.78 | 57.95 | 55.07 | 52.70 | 50.72 | 49.19 | 59.06 | **+9.12** |
| *Regularizer (Akyürek et al., 2022) | 80.37 | 74.68 | 69.39 | 65.51 | 62.38 | 59.03 | 56.36 | 53.95 | 51.73 | 63.71 | **+6.58** |
| MetaFSCIL (Chi et al., 2022) | 72.04 | 67.94 | 63.77 | 60.29 | 57.58 | 55.16 | 52.90 | 50.79 | 49.19 | 58.85 | **+9.12** |
| *C-FSCIL (Hersche et al., 2022) | 76.40 | 71.14 | 66.46 | 63.29 | 60.42 | 57.46 | 54.78 | 53.11 | 51.41 | 61.61 | **+6.90** |
| Data-free Replay (Liu et al., 2022) | 71.84 | 67.12 | 63.21 | 59.77 | 57.01 | 53.95 | 51.55 | 49.52 | 48.21 | 58.02 | **+10.10** |
| *ALICE (Peng et al., 2022) | 80.60 | 70.60 | 67.40 | 64.50 | 62.50 | 60.00 | 57.80 | 56.80 | 55.70 | 63.99 | **+2.61** |
| **\*NC-FSCIL (ours)** | **84.02** | **76.80** | **72.00** | **67.83** | **66.35** | **64.04** | **61.46** | **59.54** | **58.31** | **67.82** | |
| *Improvement over ALICE* | +3.42 | +6.20 | +4.60 | +3.33 | +3.85 | +4.04 | +3.66 | +2.74 | +2.61 | **+3.83** | |

Table 2: Performance of FSCIL in each session on CIFAR-100 and comparison with other studies. The top rows list class-incremental learning and few-shot learning results implemented by Tao et al. (2020b); Zhang et al. (2021) in the FSCIL setting. "Average Acc." is the average accuracy of all sessions. "Final Improv." calculates the improvement of our method in the last session.

| Methods | Accuracy in each session (%) ↑ | | | | | | | | | Average | Final |
| | 0 | 1 | 2 | 3 | 4 | 5 | 6 | 7 | 8 | Acc. | Improv. |
|---|---|---|---|---|---|---|---|---|---|---|---|
| iCaRL (Rebuffi et al., 2017) | 64.10 | 53.28 | 41.69 | 34.13 | 27.93 | 25.06 | 20.41 | 15.48 | 13.73 | 32.87 | **+42.38** |
| NCM (Hou et al., 2019) | 64.10 | 53.05 | 43.96 | 36.97 | 31.61 | 26.73 | 21.23 | 16.78 | 13.54 | 34.22 | **+42.57** |
| D-Cosine (Vinyals et al., 2016) | 74.55 | 67.43 | 63.63 | 59.55 | 56.11 | 53.80 | 51.68 | 49.67 | 47.68 | 58.23 | **+8.43** |
| *TOPIC (Tao et al., 2020b) | 64.10 | 55.88 | 47.07 | 45.16 | 40.11 | 36.38 | 33.96 | 31.55 | 29.37 | 42.62 | **+26.74** |
| Self-promoted (Zhu et al., 2021a) | 64.10 | 65.86 | 61.36 | 57.45 | 53.69 | 50.75 | 48.58 | 45.66 | 43.25 | 54.52 | **+12.86** |
| CEC (Zhang et al., 2021) | 73.07 | 68.88 | 65.26 | 61.19 | 58.09 | 55.57 | 53.22 | 51.34 | 49.14 | 59.53 | **+6.97** |
| DSN (Yang et al., 2022a) | 73.00 | 68.83 | 64.82 | 62.64 | 59.36 | 56.96 | 54.04 | 51.57 | 50.00 | 60.14 | **+6.11** |
| *LIMIT (Zhou et al., 2022b) | 73.81 | 72.09 | 67.87 | 63.89 | 60.70 | 57.77 | 55.67 | 53.52 | 51.23 | 61.84 | **+4.88** |
| MetaFSCIL (Chi et al., 2022) | 74.50 | 70.10 | 66.84 | 62.77 | 59.48 | 56.52 | 54.36 | 52.56 | 49.97 | 60.79 | **+6.14** |
| *C-FSCIL (Hersche et al., 2022) | 77.47 | 72.40 | 67.47 | 63.25 | 59.84 | 56.95 | 54.42 | 52.47 | 50.47 | 61.64 | **+ 5.64** |
| Data-free Replay (Liu et al., 2022) | 74.40 | 70.20 | 66.54 | 62.51 | 59.71 | 56.58 | 54.52 | 52.39 | 50.14 | 60.78 | **+5.97** |
| *ALICE (Peng et al., 2022) | 79.00 | 70.50 | 67.10 | 63.40 | 61.20 | 59.20 | 58.10 | 56.30 | 54.10 | 63.21 | **+2.01** |
| **\*NC-FSCIL (ours)** | **82.52** | **76.82** | **73.34** | **69.68** | **66.19** | **62.85** | **60.96** | **59.02** | **56.11** | **67.50** | |
| *Improvement over ALICE* | +3.52 | +6.32 | +6.24 | +6.28 | +4.99 | +3.65 | +2.86 | +2.72 | +2.01 | **+4.29** | |

## 5.2 PERFORMANCE ON BENCHMARKS

Our experiment results on minImageNet, CIFAR-100, and CUB-200 are shown in Table 1, Table 2, and Table 4 (Appendix C), respectively. We see that our method achieves the best performance in all sessions on both miniImageNet and CIFAR-100 compared with previous studies. ALICE (Peng et al., 2022) is a recent study that achieves strong performances on FSCIL. Compared with this challenging baseline, we have an improvement of 2.61% in the last session on miniImageNet, and 2.01% on CIFAR-100. We achieve an averaged accuracy improvement of more than 3.5% on both miniImageNet and CIFAR-100. Although we do not surpass ALICE in the last session on CUB-200, we still have the best average accuracy among all methods. As shown in the last rows of Table 1 and Table 2, the improvement of our method lasts and even becomes larger in the first several sessions. It indicates that our method is able to hold the superiority and relieve the forgetting of old sessions.

## 5.3 ABLATION STUDIES

We consider three models to validate the effects of ETF classifier and DR loss. All three models are based on the same framework introduced in Section 4.3 including the backbone network, the projection layer, and the memory module. The first model uses a learnable classifier and the CE

Table 3: Ablation studies on three datasets to investigate the effects of ETF classifier and DR loss. "Learnable+CE" uses a learnable classifier and the CE loss; "ETF+CE" adopts our ETF classifier with the CE loss; "ETF+DR" uses both ETF classifier and DR loss. "FINAL" refers to the accuracy of the last session; "AVERAGE" is the average accuracy of all sessions; "PD" denotes the performance drop, *i.e.,* the accuracy difference between the first and the last sessions.

| Methods | miniImageNet | | | CIFAR-100 | | | CUB-200 | | |
|---|---|---|---|---|---|---|---|---|---|
| | FINAL↑ | AVERAGE↑ | PD↓ | FINAL↑ | AVERAGE↑ | PD↓ | FINAL↑ | AVERAGE↑ | PD↓ |
| Learnable+CE | 50.04 | 61.30 | 34.53 | 52.13 | 62.68 | 30.14 | 50.38 | 59.58 | 29.19 |
| ETF+CE | 56.66 | 68.23 | 28.21 | 54.42 | 64.00 | 27.36 | 56.83 | 65.51 | 23.27 |
| ETF+DR | 58.31 | 67.82 | 25.71 | 56.11 | 67.50 | 26.41 | 59.44 | 67.28 | 21.01 |

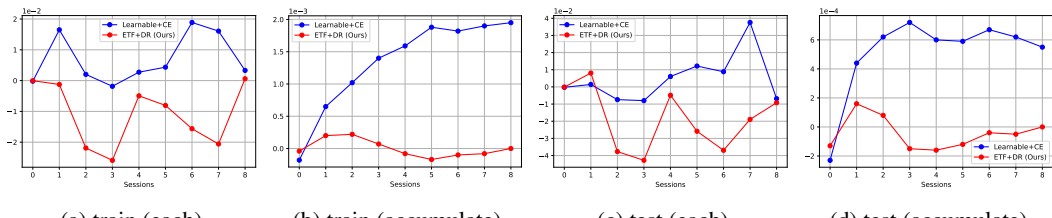

| (a) train (each) | (b) train (accumulate) | (c) test (each) | (d) test (accumulate) |
|---|---|---|---|

Figure 3: Average cosine similarity between features and classifier prototypes of different classes, *i.e.,* $\mathrm{Avg}_{k \neq k'}\{\cos \angle(\mathbf{m}_k - \mathbf{m}_G, \mathbf{w}_{k'})\}$, where $\mathbf{m}_k$ is the within-class mean of class $k$ features, $\mathbf{m}_G$ denotes the global mean, and $\mathbf{w}_{k'}$ is the classifier prototype of class $k'$. Statistics are performed among classes in each session (a and c), and all encountered classes by the current session (b and d), on train set (a and b) and test set (c and d), for models trained after each session on miniImageNet.

loss, which is the most adopted practice. The second model only replaces the classifier with our ETF classifier and also uses the CE loss. The third model corresponds to our method using both ETF classifier and DR loss. As shown in Table 3, when a fixed ETF classifier is used, the final session accuracies are significantly better, and the performance drops get much mitigated. Adopting the DR loss is able to further moderately improve the performances. It indicates that the success of our method is largely attributed to ETF classifier and DR loss, as they pre-assign a neural collapse inspired alignment and drive a model towards the fixed optimality, respectively.

## 5.4 FEATURE-CLASSIFIER STRUCTURE

We check the feature-classifier alignment instructed by neural collapse using our method and "Learnable+CE" as a comparison. As shown in Figure 3, the average cosine similarities between features and classifier prototypes of different classes, *i.e.,* $\mathrm{Avg}_{k \neq k'}\{\cos \angle(\mathbf{m}_k - \mathbf{m}_G, \mathbf{w}_{k'})\}$, of our method are consistently lower than those of the baseline. Most values of our method are negative and close to 0, which is in line with the guidance from neural collapse as derived in Eq. (10). Particularly in Figure 3b and Figure 3d, the average cosine similarities between $\mathbf{m}_k - \mathbf{m}_G$ and $\mathbf{w}_{k'}$ ($k \neq k'$) among all encountered classes increase fast with session for the baseline method, while ours keep relatively flat. It indicates that the baseline method reduces the feature-classifier margin of different classes as training incrementally, and our method enjoys a stable alignment. As shown in Figure 4 and Figure 5, we also calculate the average cosine similarities between feature and classifier of the same class, *i.e.,* $\mathrm{Avg}_k\{\cos \angle(\mathbf{m}_k - \mathbf{m}_G, \mathbf{w}_k)\}$, and the trace ratio of within-class covariance to between-class covariance, $\mathrm{tr}(\Sigma_W)/\mathrm{tr}(\Sigma_B)$. These results together support that our method better holds the feature-classifier alignment and relieves the forgetting problem.

## 6 CONCLUSION

In this paper, we propose to fix a learnable classifier as a geometric structure instructed by neural collapse for FSCIL. It pre-assigns an optimal feature-classifier alignment as a fixed target throughout incremental training, which avoids optimization conflict among sessions. Accordingly, a novel loss function that drives features towards this pre-assigned optimality is adopted **without** any regularizer. Both theoretical and empirical results support that our method is able to hold the alignment in an incremental fashion, and thus relieve the forgetting problem. In experiments of FSCIL, we achieve and even surpass the state-of-the-art performances on three datasets.

## ACKNOWLEDGMENTS

Z. Lin was supported by National Key R&D Program of China (2022ZD0160302), the major key project of PCL, China (No. PCL2021A12), the NSF China (No. 62276004), Qualcomm, and Project 2020BD006 supported by PKU-Baidu Fund.

## STATEMENTS

**Ethics Statement.** Our study does NOT involve any of the potential issues such as human subject, public health, privacy, fairness, security, *etc*. All authors of this paper confirm that they adhere to the ICLR Code of Ethics.

**Reproducibility Statement.** For our theoretical result Theorem 1, we offer the proof in Appendix A. All datasets used in this paper are public and have been cited. Please refer to Appendix B for the dataset descriptions and the implementation details of our experiments. Our source code is released at `https://github.com/NeuralCollapseApplications/FSCIL`.

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

# A APPENDIX: PROOF OF THEOREM 1

Our proof is following Yang et al. (2022b). We consider the problem in Eq. (9),

$$\min_{\mathbf{M}^{(t)}} \quad \frac{1}{N^{(t)}} \sum_{k=1}^{K^{(t)}} \sum_{i=1}^{n_k} \mathcal{L}\left(\mathbf{m}_{k,i}^{(t)}, \hat{\mathbf{W}}_{\text{ETF}}\right), \ 0 \le t \le T,$$

$$s.t. \quad \|\mathbf{m}_{k,i}^{(t)}\|^2 \le 1, \quad \forall 1 \le k \le K^{(t)}, \ 1 \le i \le n_k,$$

where $\mathbf{m}_{k,i}^{(t)} \in \mathbb{R}^d$ denotes a feature variable that belongs to the $i$-th sample of class $k$ in session $t$, $n_k$ is number of samples in class $k$, $K^{(t)}$ is number of classes in session $t$, $N^{(t)}$ is the number of samples in session $t$, *i.e.,* $N^{(t)} = \sum_{k=1}^{K^{(t)}} n_k$, and $\mathbf{M}^{(t)} \in \mathbb{R}^{d \times N^{(t)}}$ denotes a collection of $\mathbf{m}_{k,i}^{(t)}$. $\hat{\mathbf{W}}_{\text{ETF}} \in \mathbb{R}^{d \times K}$ refers to the ETF classifier for the whole label space as introduced in Section 4.1. We have $K = \sum_{t=0}^{T} K^{(t)}$ and,

$$\hat{\mathbf{w}}_k^T \hat{\mathbf{w}}_{k'} = \frac{K}{K-1} \delta_{k,k'} - \frac{1}{K-1}, \quad \forall k, k' \in [1, K], \tag{11}$$

where $\hat{\mathbf{w}}_k$ and $\hat{\mathbf{w}}_{k'}$ are two column vectors in $\hat{\mathbf{W}}_{\text{ETF}}$. From the definition of a simplex ETF in Eq. (1), we have $\hat{\mathbf{W}}_{\text{ETF}} \cdot \mathbf{1}_K = \mathbf{0}_d$, where $\mathbf{1}_K$ is an all-ones vector in $\mathbb{R}^K$, and $\mathbf{0}_d$ is an all-zeros vector in $\mathbb{R}^d$. Then we have,

$$\sum_{k=1}^{K} \hat{\mathbf{w}}_k = \mathbf{0}_d. \tag{12}$$

When $\mathcal{L}$ is the dot-regression (DR) loss in Eq. (4), it is easy to identify that $\mathcal{L} \ge 0$ and the equality holds if and only if $\hat{\mathbf{w}}_k^T \mathbf{m}_{k,i}^{(t)} = 1, \forall 0 \le t \le T, 1 \le k \le K, 1 \le i \le n_k$. Since $\|\hat{\mathbf{w}}_k\| = 1$ and $\|\mathbf{m}_{k,i}^{(t)}\|^2 \le 1$, we have $\hat{\mathbf{w}}_k^T \mathbf{m}_{k,i}^{(t)} \le 1$. The equality holds if and only if $\|\mathbf{m}_{k,i}^{(t)}\|^2 = 1$ and $\cos \angle(\hat{\mathbf{w}}_k, \mathbf{m}_{k,i}^{(t)}) = 1$. Denote $\hat{\mathbf{M}} = [\hat{\mathbf{M}}^{(0)}, \cdots, \hat{\mathbf{M}}^{(T)}] \in \mathbb{R}^{d \times \sum_{t=0}^{T} N^{(t)}}$ as the global optimality of Eq. (9) for all sessions $0 \le t \le T$. For any column vector $\hat{\mathbf{m}}_{k,i}$ in $\hat{\mathbf{M}}$, we have,

$$\|\hat{\mathbf{m}}_{k,i}\| = 1, \quad \hat{\mathbf{m}}_{k,i}^T \hat{\mathbf{w}}_{k'} = \frac{K}{K-1} \delta_{k,k'} - \frac{1}{K-1}, \quad \forall k, k' \in [1, K], \ 1 \le i \le n_k,$$

which concludes the proof for DR loss.

When $\mathcal{L}$ is the cross-entropy (CE) loss, *i.e.,*

$$\mathcal{L}\left(\mathbf{m}_{k,i}^{(t)}, \hat{\mathbf{W}}_{\text{ETF}}\right) = -\log \frac{\exp(\hat{\mathbf{w}}_k^T \mathbf{m}_{k,i}^{(t)})}{\sum_{j=1}^{K} \exp(\hat{\mathbf{w}}_j^T \mathbf{m}_{k,i}^{(t)})}, \tag{13}$$

where $0 \le t \le T, 1 \le k \le K^{(t)}$, and $1 \le i \le n_k$. Since the problem is separable among $T+1$ sessions, we only analyze the $t$-th session and omit the superscript $(t)$ for simplicity. The objective in Eq. (13) is the sum of an affine function and log-sum-exp functions. When $\hat{\mathbf{W}}_{\text{ETF}}$ is fixed, the loss is convex *w.r.t* $\mathbf{m}_{k,i}$ with convex constraints. So, we can use the KKT condition for its global optimality. Based on Eq. (9) and Eq. (13), we have the Lagrange function,

$$\tilde{\mathcal{L}} = \frac{1}{N^{(t)}} \sum_{k=1}^{K^{(t)}} \sum_{i=1}^{n_k} -\log \frac{\exp(\hat{\mathbf{w}}_k^T \mathbf{m}_{k,i})}{\sum_{j=1}^{K} \exp(\hat{\mathbf{w}}_j^T \mathbf{m}_{k,i})} + \sum_{k=1}^{K^{(t)}} \sum_{i=1}^{n_k} \lambda_{k,i}(\|\mathbf{m}_{k,i}\|^2 - 1), \tag{14}$$

where $\lambda_{k,i}$ is the Lagrange multiplier. The gradient with respect to $\mathbf{m}_{k,i}$ takes the form of:

$$\frac{\partial \tilde{\mathcal{L}}}{\partial \mathbf{m}_{k,i}} = -\frac{(1 - p_k)}{N^{(t)}} \hat{\mathbf{w}}_k + \frac{1}{N^{(t)}} \sum_{j \ne k}^{K} p_j \hat{\mathbf{w}}_j + 2\lambda_{k,i} \mathbf{m}_{k,i}, \tag{15}$$

where $1 \le i \le n_k, 1 \le k \le K^{(t)}$, and $p_j$ is the softmax probability of $\mathbf{m}_{k,i}$ for the $j$-th class, *i.e.,*

$$p_j = \frac{\exp(\hat{\mathbf{w}}_j^T \mathbf{m}_{k,i})}{\sum_{j'=1}^{K} \exp(\hat{\mathbf{w}}_{j'}^T \mathbf{m}_{k,i})}. \tag{16}$$

Since $\|\hat{\mathbf{w}}_k\| = 1$ and $\|\mathbf{m}_{k,i}\| \leq 1$, we have $0 < p_k < 1, \forall 1 \leq k \leq K$.

We now solve the equation $\frac{\partial \tilde{\mathcal{L}}}{\partial \mathbf{m}_{k,i}} = 0$. Assume that $\lambda_{k,i} = 0$, and then we have,

$$\sum_{j \neq k}^{K} p_j \hat{\mathbf{w}}_j = (1 - p_k)\hat{\mathbf{w}}_k. \tag{17}$$

Since $1 - p_k = \sum_{j \neq k}^{K} p_j$ and Eq. (11), multiplying $\hat{\mathbf{w}}_k$ by both sides of Eq. (17), we have,

$$\frac{K}{K-1}(1 - p_k) = 0, \tag{18}$$

which contradicts with $0 < p_k < 1$. Then we have the other case $\lambda_{k,i} > 0$. Based on the KKT condition, the global optimality $\hat{\mathbf{m}}_{k,i}$ satisfies that

$$\|\hat{\mathbf{m}}_{k,i}\|^2 = 1. \tag{19}$$

The equation $\frac{\partial \tilde{\mathcal{L}}}{\partial \hat{\mathbf{m}}_{k,i}} = 0$ leads to:

$$\sum_{j \neq k}^{K} p_j(\hat{\mathbf{w}}_j - \hat{\mathbf{w}}_k) + 2N^{(t)}\lambda_{k,i}\hat{\mathbf{m}}_{k,i} = 0. \tag{20}$$

Based on Eq. (11), for any $j' \neq k$, multiplying $\hat{\mathbf{w}}_{j'}$ by both sides of Eq. (20), we have,

$$p_{j'}\frac{K}{K-1} + 2N^{(t)}\lambda_{k,i}\hat{\mathbf{m}}_{k,i}^T\hat{\mathbf{w}}_{j'} = 0. \tag{21}$$

Since $\forall k \in [1, K]$, $p_k > 0$, we have $\hat{\mathbf{m}}_{k,i}^T\hat{\mathbf{w}}_{j'} < 0$. Then for any $j_1, j_2 \neq k$,

$$\frac{p_{j_1}}{p_{j_2}} = \frac{\exp(\hat{\mathbf{w}}_{j_1}^T\hat{\mathbf{m}}_{k,i})}{\exp(\hat{\mathbf{w}}_{j_2}^T\hat{\mathbf{m}}_{k,i})} = \frac{\hat{\mathbf{w}}_{j_1}^T\hat{\mathbf{m}}_{k,i}}{\hat{\mathbf{w}}_{j_2}^T\hat{\mathbf{m}}_{k,i}}. \tag{22}$$

The function $f(x) = \exp(x)/x$ is monotonically increasing when $x < 1$. So, Eq. (22) indicates that

$$p_{j_1} = p_{j_2}, \ \hat{\mathbf{w}}_{j_1}^T\hat{\mathbf{m}}_{k,i} = \hat{\mathbf{w}}_{j_2}^T\hat{\mathbf{m}}_{k,i}, \ \forall j_1, j_2 \neq k, \tag{23}$$

and

$$p_j = \frac{1 - p_k}{K - 1} = -\frac{2N^{(t)}\lambda_{k,i}\hat{\mathbf{m}}_{k,i}^T\hat{\mathbf{w}}_j(K-1)}{K}, \ \forall j \neq k. \tag{24}$$

Multiplying $\hat{\mathbf{w}}_k$ by both sides of Eq. (20), we have,

$$-\frac{K}{K-1}(1 - p_k) + 2N^{(t)}\lambda_{k,i}\hat{\mathbf{m}}_{k,i}^T\hat{\mathbf{w}}_k = 0. \tag{25}$$

Combing Eq. (24) and Eq. (25), we have,

$$\hat{\mathbf{m}}_{k,i}^T\hat{\mathbf{w}}_j(K-1) + \hat{\mathbf{m}}_{k,i}^T\hat{\mathbf{w}}_k = 0, \ \forall j \neq k. \tag{26}$$

Based on $p_j = \frac{1-p_k}{K-1}, \forall j \neq k$, and Eq. (12), we can rewrite Eq. (20) as:

$$-\frac{(1 - p_k)K}{K-1}\hat{\mathbf{w}}_k + 2N^{(t)}\lambda_{k,i}\hat{\mathbf{m}}_{k,i} = 0, \tag{27}$$

which means that $\hat{\mathbf{m}}_{k,i}$ is aligned with $\hat{\mathbf{w}}_k$, i.e., $\cos\angle(\hat{\mathbf{m}}_{k,i}, \hat{\mathbf{w}}_k) = 1$. Given that $\|\hat{\mathbf{w}}_k\| = 1$ and $\|\hat{\mathbf{m}}_{k,i}\| = 1$ (Eq. (19)), we have,

$$\hat{\mathbf{m}}_{k,i}^T\hat{\mathbf{w}}_k = 1,$$

and Eq. (26) leads to:

$$\hat{\mathbf{m}}_{k,i}^T\hat{\mathbf{w}}_j = -\frac{1}{K-1}, \ \forall j \neq k.$$

Therefore, for any column vector $\hat{\mathbf{m}}_{k,i}$ in $\hat{\mathbf{M}}$, we have,

$$\|\hat{\mathbf{m}}_{k,i}\| = 1, \ \hat{\mathbf{m}}_{k,i}^T\hat{\mathbf{w}}_{k'} = \frac{K}{K-1}\delta_{k,k'} - \frac{1}{K-1}, \ \forall k, k' \in [1, K], \ 1 \leq i \leq n_k,$$

which concludes the proof for CE loss. $\qquad \square$

Table 4: Performance of FSCIL in each session on CUB-200 and comparison with other studies. The top rows list class-incremental learning and few-shot learning results implemented by Tao et al. (2020b); Zhang et al. (2021); Liu et al. (2022); Zhou et al. (2022a) in the FSCIL setting. "Average Acc." is the average accuracy of all sessions. "Final Improv." calculates the improvement of our method in the last session.

| Methods | Accuracy in each session (%) ↑ | | | | | | | | | | | Average | Final |
| | 0 | 1 | 2 | 3 | 4 | 5 | 6 | 7 | 8 | 9 | 10 | Acc. | Improv. |
|---|---|---|---|---|---|---|---|---|---|---|---|---|---|
| iCaRL (Rebuffi et al., 2017) | 68.68 | 52.65 | 48.61 | 44.16 | 36.62 | 29.52 | 27.83 | 26.26 | 24.01 | 23.89 | 21.16 | 36.67 | **+38.28** |
| EEIL (Castro et al., 2018) | 68.68 | 53.63 | 47.91 | 44.20 | 36.30 | 27.46 | 25.93 | 24.70 | 23.95 | 24.13 | 22.11 | 36.27 | **+37.33** |
| NCM (Hou et al., 2019) | 68.68 | 57.12 | 44.21 | 28.78 | 26.71 | 25.66 | 24.62 | 21.52 | 20.12 | 20.06 | 19.87 | 32.49 | **+39.57** |
| Fixed classifier (Pernici et al., 2021) | 68.47 | 51.00 | 45.42 | 40.76 | 35.90 | 33.18 | 27.23 | 24.24 | 21.18 | 17.34 | 16.20 | 34.63 | **+43.24** |
| D-NegCosine (Liu et al., 2020) | 74.96 | 70.57 | 66.62 | 61.32 | 60.09 | 56.06 | 55.03 | 52.78 | 51.50 | 50.08 | 48.47 | 58.86 | **+10.97** |
| D-DeepEMD (Zhang et al., 2020) | 75.35 | 70.69 | 66.68 | 62.34 | 59.76 | 56.54 | 54.61 | 52.52 | 50.73 | 49.20 | 47.60 | 58.73 | **+11.84** |
| D-Cosine (Vinyals et al., 2016) | 75.52 | 70.95 | 66.46 | 61.20 | 60.86 | 56.88 | 55.40 | 53.49 | 51.94 | 50.93 | 49.31 | 59.36 | **+10.13** |
| DeepInv (Yin et al., 2020) | 75.90 | 70.21 | 65.36 | 60.14 | 58.79 | 55.88 | 53.21 | 51.27 | 49.38 | 47.11 | 45.67 | 57.54 | **+13.77** |
| TOPIC (Tao et al., 2020b) | 68.68 | 62.49 | 54.81 | 49.99 | 45.25 | 41.40 | 38.35 | 35.36 | 32.22 | 28.31 | 26.28 | 43.92 | **+33.16** |
| IDLVQ (Chen & Lee, 2021) | 77.37 | 74.72 | 70.28 | 67.13 | 65.34 | 63.52 | 62.10 | 61.54 | 59.04 | 58.68 | 57.81 | 65.23 | **+1.63** |
| SPPR (Zhu et al., 2021a) | 68.68 | 61.85 | 57.43 | 52.68 | 50.19 | 46.88 | 44.65 | 43.07 | 40.17 | 39.63 | 37.33 | 49.32 | **+22.11** |
| Cheraghian et al. (2021b) | 68.78 | 59.37 | 59.32 | 54.96 | 52.58 | 49.81 | 48.09 | 46.32 | 44.33 | 43.43 | 43.23 | 51.84 | **+16.21** |
| CEC (Zhang et al., 2021) | 75.85 | 71.94 | 68.50 | 63.50 | 62.43 | 58.27 | 57.73 | 55.81 | 54.83 | 53.52 | 52.28 | 61.33 | **+7.16** |
| LIMIT (Zhou et al., 2022b) | 76.32 | 74.18 | 72.68 | 69.19 | **68.79** | **65.64** | 63.57 | 62.69 | **61.47** | 60.44 | 58.45 | 66.67 | **+0.99** |
| MgSvF (Zhao et al., 2021) | 72.29 | 70.53 | 67.00 | 64.92 | 62.67 | 61.89 | 59.63 | 59.15 | 57.73 | 55.92 | 54.33 | 62.37 | **+5.11** |
| MetaFSCIL (Chi et al., 2022) | 75.9 | 72.41 | 68.78 | 64.78 | 62.96 | 59.99 | 58.3 | 56.85 | 54.78 | 53.82 | 52.64 | 61.93 | **+6.8** |
| FACT (Zhou et al., 2022a) | 75.90 | 73.23 | 70.84 | 66.13 | 65.56 | 62.15 | 61.74 | 59.83 | 58.41 | 57.89 | 56.94 | 64.42 | **+2.5** |
| Data-free replay (Liu et al., 2022) | 75.90 | 72.14 | 68.64 | 63.76 | 62.58 | 59.11 | 57.82 | 55.89 | 54.92 | 53.58 | 52.39 | 61.52 | **+7.05** |
| ALICE (Peng et al., 2022) | 77.40 | 72.70 | 70.60 | 67.20 | 65.90 | 63.40 | 62.90 | 61.90 | 60.50 | **60.60** | **60.10** | 65.75 | -0.66 |
| **NC-FSCIL (ours)** | **80.45** | **75.98** | **72.30** | **70.28** | 68.17 | 65.16 | **64.43** | **63.25** | 60.66 | 60.01 | 59.44 | **67.28** | |

Table 5: A comparison of backbone networks used in different studies.

| Methods | miniImageNet | CIFAR-100 | CUB-200 |
|---|---|---|---|
| TOPIC (Tao et al., 2020b) | ResNet-18 | ResNet-18 | ResNet-18 |
| CEC (Zhang et al., 2021) | ResNet-18 | ResNet-20 | ResNet-18 |
| CFSCIL (Hersche et al., 2022) | ResNet-12 | ResNet-12 | - |
| LIMIT (Zhou et al., 2022b) | ResNet-18 | ResNet-20 | ResNet-18 |
| ALICE (Peng et al., 2022) | ResNet-18 | ResNet-18 | ResNet-18 |
| NC-FSCIL (ours) | ResNet-12 | ResNet-12 | ResNet-18 |

# B  APPENDIX: IMPLEMENTATION DETAILS

**Datasets.** We conduct our experiments on three FSCIL benchmark datasets including miniImageNet (Russakovsky et al., 2015), CIFAR-100 (Krizhevsky et al., 2009), and CUB-200 (Wah et al., 2011). miniImageNet is a variant of ImageNet with an image size of $84 \times 84$. It has 100 classes with each class containing 500 images for training and 100 images for testing. CIFAR-100 has the same number of classes and images, and the image size is $32 \times 32$. CUB-200 is a dataset for fine-grained image classification containing 11,788 images of 200 classes in a resolution of $224 \times 224$. There are 5,994 images for training and 5,794 images for testing. We follow the standard experimental settings in FSCIL (Tao et al., 2020b; Zhang et al., 2021). For both miniImageNet and CIFAR-100, the base session ($t = 0$) contains 60 classes, and a 5-way 5-shot (5 classes and 5 images per class) problem is adopted for each of the 8 incremental sessions ($1 \leq t \leq 8$). For CUB-200, 100 classes are used in the base session, and there are 10 incremental sessions, each of which is 10-way 5-shot.

**Architectures.** Prior studies widely adopt ResNet-12, ResNet-18, and ResNet-20 (He et al., 2016) for FSCIL experiments. As shown in Table 5, we compare the backbone networks used in different studies. For miniImageNet and CIFAR-100, we use ResNet-12 following Hersche et al. (2022). For CUB-200, we use ResNet-18 (pre-trained on ImageNet) following other studies. We adopt a two-layer MLP block as the projection layer following the practice in Peng et al. (2022).

**Training Details.** We adopt the standard data pre-processing and augmentation schemes including random resizing, random flipping, and color jittering (Tao et al., 2020b; Zhang et al., 2021; Peng et al., 2022). We train all models with a batchsize of 512 in the base session, and a batchsize of 64 (containing new session data and intermediate features in the memory) in each incremental session. On miniImageNet, we train for 500 epochs in the base session, and 100-170 iterations in each incremental session. The initial learning rate is 0.25 for base session, and 0.025 for incremental sessions. On CIFAR-100, we train for 200 epochs in the base session, and 50-200 iterations in each incremental session. The initial learning rate is 0.25 for both base and incremental sessions. On CUB-200, we train for 80 epochs in the base session, and 105-150 iterations in each incremental session. The initial learning rates are 0.025 and 0.05 for base session and incremental sessions, respectively. In all experiments, we adopt a cosine annealing strategy for learning rate, and use SGD with momentum as optimizer. Our code will be publicly available in the final version.

## C  APPENDIX: MORE RESULTS

Our experimental result on CUB-200 is shown in Table 4. We achieve a better accuracy in the last session than most of the baseline methods. Although we do not surpass ALICE in the last session on CUB-200, we still have the best average accuracy among all methods.

We also visualize the average cosine similarities between feature and classifier of the same class, *i.e.,* $\mathrm{Avg}_k\{\cos\angle(\mathbf{m}_k - \mathbf{m}_G, \mathbf{w}_k)\}$ and the trace ratio of within-class covariance to between-class covariance, $\mathrm{tr}(\Sigma_W)/\mathrm{tr}(\Sigma_B)$.

A higher average $\cos\angle(\mathbf{m}_k - \mathbf{m}_G, \mathbf{w}_k)$ indicates that feature centers are more closely aligned with their corresponding classifier prototypes of the same class. As shown in Figure 4, the values of our method are consistently higher than those of the baseline method. Figure 4a and Figure 4d reveal that our method has a better feature-classifier alignment in each session of the incremental training on both train and test sets. When we measure on all the encountered classes by each session in Figure 4b and Figure 4e, the metric for our method does not change obviously after the 4-th session on train set, while the metric for the baseline method keeps decreasing as training incrementally. Especially for the base session classes, our method is able to keep the metric stable on both train and test sets after the decline of the first 3-4 sessions, as shown in Figure 4c and Figure 4f. As a comparison, the baseline method cannot mitigate the deterioration. Given that the base session has the most classes, the performance on base session classes largely decides the final accuracy in the last session for FSCIL. Therefore, the superiority of our method can be attributed to our ability of keeping the feature-classifier alignment well for base session classes.

The within-class covariance $\Sigma_W$ and the between-class covariance $\Sigma_B$ are defined as:

$$\Sigma_W = \mathrm{Avg}_k\{\Sigma_W^{(k)}\}, \quad \Sigma_W^{(k)} = \mathrm{Avg}_i\{(\mathbf{m}_{k,i} - \mathbf{m}_k)(\mathbf{m}_{k,i} - \mathbf{m}_k)^T\},$$

and

$$\Sigma_B = \mathrm{Avg}_k\{(\mathbf{m}_k - \mathbf{m}_G)(\mathbf{m}_k - \mathbf{m}_G)^T\},$$

where $\mathbf{m}_{k,i}$ is the feature of sample $i$ in class $k$, $\mathbf{m}_k$ is the within-class mean of class $k$ features, and $\mathbf{m}_G$ denotes the global mean of all features. A lower within-class variation with a higher between-class variation corresponds to a better Fisher Discriminant Ratio. As shown in Figure 5, we compare the trace ratio of within-class covariance to between-class covariance between our method and the baseline method. We observe similar patterns to Figure 4. Concretely, the trace ratio metric of our method is consistently lower that of baseline. For the base session classes, the metric of our method increases more mildly, which corroborates our ability of maintaining the performance on the old classes, and is in line with the indications from Figure 3 and Figure 4.

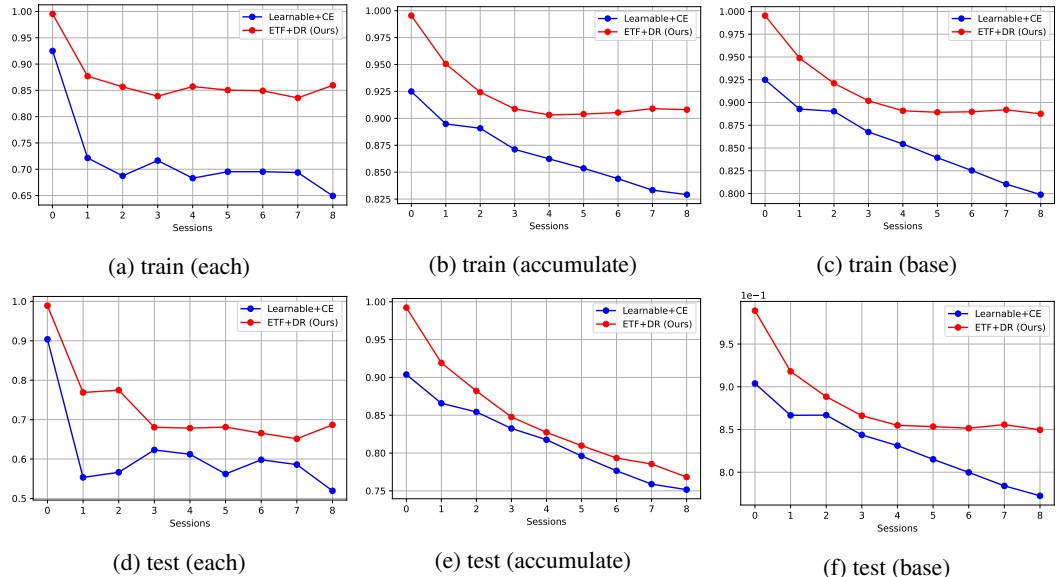

Figure 4: Average cosine similarity between features and classifier prototypes of the same class, *i.e.,* $\mathrm{Avg}_k\{\cos\angle(\mathbf{m}_k - \mathbf{m}_G, \mathbf{w}_k)\}$, where $\mathbf{m}_k$ is the within-class mean of class $k$ features, $\mathbf{m}_G$ denotes the global mean, and $\mathbf{w}_k$ is the classifier prototype of class $k$. Statistics are performed among classes in each session (a and d), all encountered classes by the current session (b and e), and only the base session classes (c and f), on train set (a, b, c) and test set (d, e, f), for models trained after each session on miniImageNet.

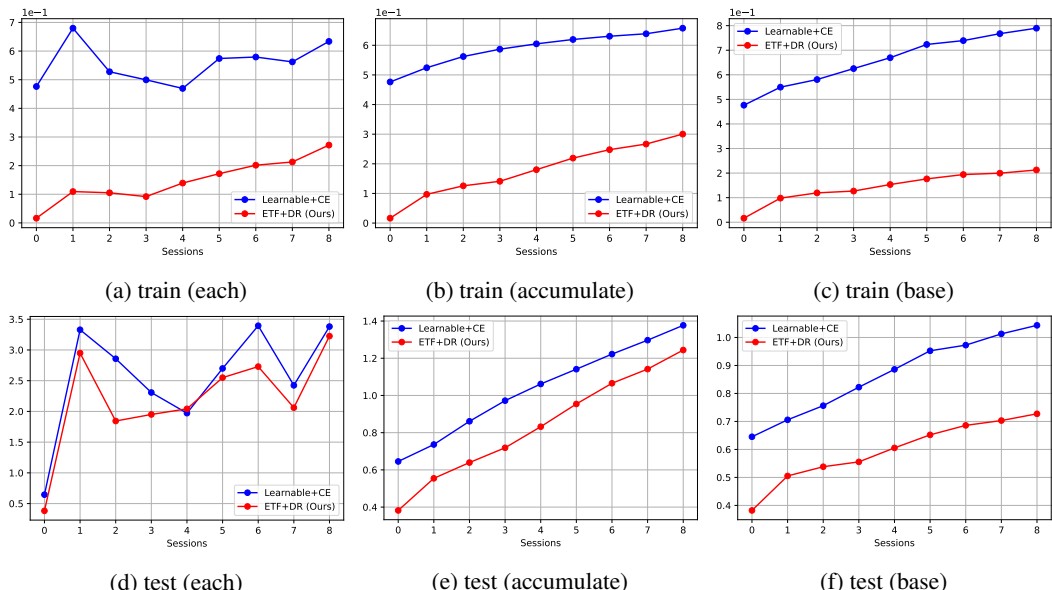

Figure 5: Trace ratio of within-class covariance to between-class covariance, *i.e.,* $\mathrm{tr}(\Sigma_W)/\mathrm{tr}(\Sigma_B)$, where $\Sigma_W$ is the within-class covariance, and $\Sigma_B$ denotes the between-class covariance. Statistics are performed among classes in each session (a and d), all encountered classes by the current session (b and e), and only the base session classes (c and f), on train set (a, b, c) and test set (d, e, f), for models trained after each session on miniImageNet.

