# OpenReview forum: "Neural Collapse Inspired Feature-Classifier Alignment for Few-Shot Class-Incremental Learning"
_ICLR.cc/2023/Conference — ICLR 2023 notable top 25%_

### Official Review · Reviewer_NNii · 2022-10-23

**Confidence:** 4
**Clarity, Quality, Novelty And Reproducibility:** See the Strength and Weaknesses.
**Correctness:** 2
**Technical Novelty And Significance:** 2
**Empirical Novelty And Significance:** 3
**Recommendation:** 6

**Strength And Weaknesses:**

Strength:

The proposed framework that combines neural collapse inspired framework in the few-shot class incremental learning is interesting and inspiring. I also appreciate the extensive experiments (ablation studies and extensions) conducted by the authors.

(1) The proposed neural collapse inspired framework on few-shot incremental learning seems novel.  The pre-assigned classifier could avoid optimization conﬂict in each session in the training stage. Its effectiveness was successfully proved by experiments and theoretical analysis. This is the biggest contribution of this paper.   Although adding an ETF classifier is not so novel, its effectiveness was shown by the experiments.

(2) The experiments in the supplementary material are comprehensive and detailed.

Weakness:

Major concerns:

(1) The writing needs to be improved (not for the problem of English expression).  e.g.,  The last sentence of the first paragraph. What’s the importance and difficulty of the FSCIL?  The first sentence of the first contribution is completely uninformative and does not show their contribution at all.

(2) In the second paragraph, "Besides, as shown in Figure 1 (a), there will be a misalignment between the adjusted classiﬁer and the ﬁxed features of old classes." How did you know the misalignment between the adjusted classiﬁer and the ﬁxed features of old classes?  and in Figure 1 (a), how did you know the old features and new classes are close?

(3) Apart from the fact that the motivation of neural collapse inspired framework is novel, the novelty of this paper still seems incremental.
They only use Yang et al. method for the few-shot incremental learning.  Even the written in sec3.2 is similar to Yang et al. Could you please explain the technical innovation of your approach versus the Yang et al. approach other than in FSCIL?

(4) Since this paper pre-assigned the classifier at the beginning of all classes, even the tested classes,   but in real life, we cannot know how many classes will be tested during the test stage. So how does this method work when there is no way to know the number of classes to test?

(5) It would be better to separate the comparisons between memory-based and non-memory-based methods in Table 1,2.









**Summary Of The Paper:**

Inspired by Neural collapse, this paper proposed to ﬁx a learnable classiﬁer as a geometric structure for few-shot incremental learning.
The pre-assigned classifier could avoid optimization conﬂict among sessions in the training stage. They also provided a detailed theoretical analysis to show their method could hold the neural collapse optimality.  Experiments on the three datasets achieve the best performance compared with previous methods.

**Summary Of The Review:**

I believe this paper is slightly below the acceptance threshold.


The paper presents a novel framework that inspired neural collapse for the FSCIL.  and quantitative experiments show that the proposed method outperforms baseline methods.   But, the overall novelty is incremental. (See the weakness.)

But if the authors can explain well the weakness I raised, I will consider raising my score.


============== Post Rebuttal Update

Sorry for the late response. I thank the authors for their diligent efforts in the rebuttal. It was nice to see that the authors could give more detailed intuitions to improve their work and make relevant changes to their paper. Also, based on the other reviewer's comments and the responded of the authors, I have increased my score based on the author's responses.

---

> ### Author Response · Authors · 2022-11-16
> **Response to Reviewer NNii (1)**
>
> Thank you for recognizing the contributions of our paper and your valuable feedback.
>
> > (1)	The writing, particularly the last sentence of the first paragraph, needs to be improved. What’s the importance and difficulty of the FSCIL? The first sentence of the first contribution is uninformative and does not show the contribution at all.
>
> Many thanks for your suggestion. We have revised our paper according to your comments. We quote the corresponding contents here. Please see the current paper for our revision, which is highlighted.
>
> - What’s the importance and difficulty of the FSCIL?
>
> “Learning incrementally and learning with few-shot labeled samples are common in the real-world implementations, and in many applications, such as robotics, the two demands emerge simultaneously.” We use this sentence to show the necessity and importance of FSCIL.
>
> “It poses the notorious catastrophic forgetting problem because the novel sessions have no access to the data of the previous sessions.” We use this sentence to show the difficulty of FSCIL.
>
> We have re-organized the first three paragraphs to make the logic clearer. In the revised paper, the first paragraph draws forth the task of FSCIL from its necessity and importance. The second paragraph briefly introduces FSCIL, including its problem setting, its difference from few-shot learning and incremental learning, and its difficulty. We use the sentence “Due to the importance and difficulty, FSCIL has attracted much research attention” to transit into the third paragraph, where we briefly summarize the existing studies and the problem they suffer from, i.e., misalignment dilemma, which naturally connects with the question and motivation posed by our study in the later paragraphs.
>
> - The first contribution.
>
> We have revised the first contribution. Please see the current paper for the revision. We quote it here for your reference.
>
> First contribution: “To relieve the misalignment dilemma in FSCIL, we propose to pre-assign an optimal alignment inspired by neural collapse as a fixed target throughout the incremental learning. Our model is trained towards the same optimality to avoid optimization conflict among sessions.”

---

> > ### Author Response · Authors · 2022-11-16
> > **Response to Reviewer NNii (2)**
> >
> > > (2)	How did the authors know the misalignment between the adjusted classifier and fixed features of old classes? In Fig 1(a), how did the authors know the old features and new classes are close?
> >
> > - Old features and new classes are close.
> >
> > Please note that what we claim is “the newly introduced prototypes (for new classes) may lie close to the old-class prototypes”. It is prototypes instead of features. Assume that the base session has $K_0$ classes. After the training of base session (the general training without any regularization), the prototypes of the $K_0$ classes tend to maximally divide the feature space, as shown in the leftmost figure in Fig. 1(a). In the later sessions, the newly-learned prototypes have to locate at the margin between the old-class prototypes. In this way, the prototypes of new classes may be close to that of the old classes. Dividing the feature space incrementally cannot ensure an optimal assignment of prototypes for all classes. By contrast, we pre-assign the prototypes for all classes according to the instruction of neural collapse. As a result, all $K$ classes in this task equally and maximally divide the feature space, i.e., any two classes have the same cosine similarity ($-\frac{1}{K-1}$), and the cosine similarity corresponds to the largest possible equiangular separation of $K$ classes in a high dimension space.
> >
> > Actually, the study FACT [1] has a similar motivation to ours. They propose to reserve the feature space for later classes in each session’s training, so the novel-class prototypes can be located with a distance with the old-class ones. But their method relies on sophisticated loss functions and regularizers for this purpose, which cannot ensure to attain an ideal structure such as the one instructed by neural collapse as adopted in our method.
> >
> > - Misalignment between the adjusted classifier and fixed feature of old classes.
> >
> > As responded to the last point above, the newly-learned prototypes for novel classes may lie close to the old-class ones. To this end, many studies introduce regularizers to shift the old-class prototypes to make them separated from the novel-class ones with enough distance, e.g. Eq. (7) in C-FSCIL [2]. However, each incremental session only has few-shot data. Training the backbone network on few-shot data will cause severe overfitting. As widely adopted, the backbone network is fixed after the base session training, so the old-class features are fixed. As a result, there will be a misalignment between the shifted prototypes and the fixed backbone features for old classes. That is why many studies will use another regularizer to prevent the old-class prototypes from shifting far away from their original position, e.g., Eq. (9) in C-FSCIL [2].
> >
> > We will make the corresponding descriptions about the two points above clearer in the final version of our paper.

---

> > > ### Author Response · Authors · 2022-11-16
> > > **Response to Reviewer NNii (3)**
> > >
> > > > (3)	The authors only use Yang et al. method for FSCIL. Explain the technical innovation of the approach versus Yang et al. other than in FSCIL.
> > >
> > > The use of ETF classifier and the DR loss, is indeed following Yang et al. [3] as described in our paper. We never claim that we originally design ETF classifier and DR loss in our contributions. Our contributions focus on their application into FSCIL and the resulting benefits to FSCIL. We cite their paper frequently at the corresponding contents. Yang et al. [3] show that using the ETF classifier helps to induce neural collapse in imbalanced classification, and proves the advantage of DR loss over CE loss. In technical methodology, our analysis and implementation are in the context of FSCIL. Concretely, the number of classes are specified by the whole label space in the FSCIL task. We also introduce the projection layer and the memory of feature means. The pipeline in Figure 2 is significantly different from Yang et al. [3] because we focus on different tasks. **More importantly, the insights from neural collapse brought by our work are novel to the FSCIL area.**
> > >
> > > Besides, we would like to highlight the following views.
> > >
> > > First, the successful application of neural collapse theory/inspiration into the task of FSCIL should be a significant contribution. Yang et al., [3] try to induce neural collapse in imbalanced classification. Different from their work, we show that learning incrementally would also break the neural collapse state because the feature space is divided incrementally in existing studies, so we try to induce neural collapse in FSCIL. We think our work is another important application of neural collapse and brings novel insights to FSCIL. The motivation from neural collapse and the use of ETF classifier to induce the ETF structure fit well for solving the misalignment dilemma in FSCIL. As shown in Tab. 1 and 2, our simple pipeline already achieves remarkable improvements, which supports our motivation well. We do not think it is necessary to extra develop some new module/design just to intentionally make the pipeline look more complex and novel.
> > >
> > > Second, designing a totally new method should not be the only criterion to judge a paper. Many studies apply a method of one area to successfully solve the problem of another one. We would like to use non-local network [5] as an example. Inspired by the non-local mean algorithm [6] that is initially proposed for image denoising, non-local network [5] applies the non-local mean formulation into visual recognition for long-range dependency, and achieves significant improvements. Similarly, neural collapse should be a promising perspective for not only imbalanced classification, but also FSCIL. We are the first to bring the solutions from neural collapse into FSCIL, and effectively mitigate the misalignment dilemma in existing studies. So, compared with the method itself, the insights behind the method adopted seem to be more crucial for solving a problem.
> > >
> > > Finally, Sec 3.2 is used to introduce the background of neural collapse. The definition of ETF and the properties of neural collapse are preliminary knowledges and are invariant. So, this part has to be similar to the introduction of neural collapse in the other studies, such as Yang et al. [3] and Ji et al. [4].
> > >
> > > As a conclusion, we hold that our work is an advancement to FSCIL, and our contributions should not be denied just because we apply a method from another area into FSCIL. We hope our response could alleviate your concern about the methodology.

---

> > > > ### Author Response · Authors · 2022-11-16
> > > > **Response to Reviewer NNii (4)**
> > > >
> > > > > (4)	In real life, we cannot know how many classes will be tested during the test stage. How does this method work when there is no way to know the number of classes to test?
> > > >
> > > > Thanks for your interesting question. Please note that in FSCIL the number of classes to test after each session is all the classes that have been encountered (trained) by the current session. So, we can know the number of classes to test just seeing how many classes does the model have been trained on. We guess what you actually care is when there is no way to know the number of total classes.
> > > >
> > > > Indeed, our method pre-assigns the classifier for the whole label space of the incremental task so needs to know the number of total classes. But when we cannot know the total classes, our method will also work well. In this case, we can initialize the classifier for a very large number of classes, denoted as $K_M$, such that the real classes to be encountered, denoted as $K$, will be smaller than $K_M$. The $K_M$ prototypes will be also equiangular. The difference is that the pair-wise cosine similarity is increased from $-\frac{1}{K-1}$ to $-\frac{1}{K_M -1}$. But both the two similarities are near 0 and near to be orthogonal. So, it almost does not affect the learning performance.
> > > >
> > > > In order to support the effectiveness of this practice, we make an experiment here. On miniImageNet (100 total calsses), we initialize the ETF classifier for 500 classes, and only randomly use 100 prototypes out of the 500 prototypes in the incremental training. The results are compared as follows.
> > > >
> > > > | Methods | 0 | 1 | 2 | 3 | 4 | 5 | 6 | 7 | 8 |
> > > > | --- | --- | --- | --- | --- | --- | --- | --- | --- | --- |
> > > > | NC-FSCIL (100 classes) | 84.02 | 76.80 | 72.00 | 67.83 | 66.35 | 64.04 | 61.46 | 59.54 | 58.31 |
> > > > | NC-FSCIL (500 classes) | 84.20 | 76.63 | 72.37 | 68.27 | 66.83 | 64.68 | 61.52 | 60.21 | 58.42 |
> > > >
> > > > It is shown that the result of 500 classes is very close to the 100-class result adopted in our paper.
> > > >
> > > > > (5)	It is better to separate the comparisons between memory-based and non-memory-based method in Table 1 and Table 2.
> > > >
> > > > Thanks for your suggestion. We have revised Table 1 and Table 2. Methods that need a memory of feature mean for training or inference are marked by *. We also summarize them here for your reference.
> > > >
> > > > - Save feature mean for training or inference: NC-FSCIL (Ours), TOPIC, IDLVQ, Regularizer, C-FSCIL, Sematic-aware KD, LIMIT, FACT, ALICE;
> > > > - No memory of feature mean: Self-promoted, CEC, MetaFSCIL, Data-Free;
> > > >
> > > > Note that saving the feature mean of each class has been popular in FSCIL and few-shot learning since ProtoNet [7]. It does not consume much memory because only one vector is saved for each class.
> > > >
> > > > ### References:
> > > > [1] Zhou et al., Forward compatible few-shot class-incremental learning, CVPR 2022.
> > > >
> > > > [2] Hersche et al., Constrained few-shot class-incremental learning, CVPR 2022.
> > > >
> > > > [3] Yang et al., Do we really need a learnable classifier at the end of deep neural network?, NeurIPS 2022.
> > > >
> > > > [4] Ji et al., An unconstrained layer-peeled perspective on neural collapse, ICLR 2022.
> > > >
> > > > [5] Wang et al., Non-local Neural Networks, CVPR 2018.
> > > >
> > > > [6] A. Buades et al., A non-local algorithm for image denoising. CVPR 2005.
> > > >
> > > > [7] Snell et al., Prototypical Networks for Few-shot Learning, NeurIPS 2017.

---

> > > > > ### Author Response · Authors · 2022-11-30
> > > > > **Thanks again**
> > > > >
> > > > > Dear Reviewer NNii:
> > > > >
> > > > > We would like to thank you again for the efforts in reviewing and the valuable comments. We have tried our best to address your concerns in the response and revise our paper accordingly. We sincerely hope that you could consider raising your score based on our response. We look forward to any discussion opportunity. Please let us know if there is more question or any concern is still remaining.
> > > > >
> > > > > Best regards,
> > > > >
> > > > > Authors

---

### Official Review · Reviewer_Smvh · 2022-10-24

**Confidence:** 3
**Clarity, Quality, Novelty And Reproducibility:** Overall, the paper is well written. T…
**Correctness:** 4
**Technical Novelty And Significance:** 3
**Empirical Novelty And Significance:** 2
**Recommendation:** 8

**Strength And Weaknesses:**

Strength:

The proposed method is an innovative application of Neural Collapse to incremental learning. The method seems to be effective in the experiments shown.





**Summary Of The Paper:**

This paper proposed a new loss function for incremental learning, inspired by the inductive bias of neural collapse. The basic idea is to preallocate a sufficiently large Equiangular Tight Frame (ETF),  so that features from different classes (old and new) will be maximally separated from each other. At the same time, use remembered features from the old classes to fix the representations of the old classes.

The authors empirically show the effectiveness of the proposed method.

**Summary Of The Review:**

Overall, the authors proposed a simple yet effective solution to catastrophic forgetting in incremental learning. I hope that the authors can publish the code for the sake of reproducibility.

---

> ### Author Response · Authors · 2022-11-16
> **Response to Reviewer Smvh**
>
> Thank you for recognizing the contributions of our paper and your valuable feedback.
>
> > (1)	The reviewer hope that the authors can publish the code for the sake of reproducibility.
>
> Yes, we will of course release our project repo in the final paper, including the code, the trained models, the training logs, and the docker file for re-producibility.

---

### Official Review · Reviewer_mQxd · 2022-10-25

**Confidence:** 2
**Correctness:** 2
**Technical Novelty And Significance:** 2
**Empirical Novelty And Significance:** 1
**Recommendation:** 6

**Clarity, Quality, Novelty And Reproducibility:**

The clarity of the ``neural collapse” formulation could be improved. The originality is good. Using fixed prototypes to equally divide the feature space is a novel idea. There is no code provided, such that the re-producibility of the submission is difficult to be evaluated.

**Strength And Weaknesses:**

Strength:

The idea is novel and interesting. The proposed FSCIL method assigns neural collapse inspired feature-classifier alignment as a target and trains a model towards the same optimization in each session to avoid class confusion and conflict. The performance is compare to, if not outperforms, the recently published methods.

Weakness:

The formulation of neural collapse is not that strict. In my view, the ``neural collapse” appears more like a class confusion in a feature space. The neural collapse, as illustrated in Fig. 1, is also built on the assumption that the feature space is fixed. While as is known, the feature space would change significantly during the model training (either base training or finetuning) procedure. I would like the authors to clarify this point during rebuttal.

The authors claimed that the proposed approach achieved an average 3.5\% accuracy improvement over the STOA. But I note in the first session (session 0) in Table 1, the proposed approach has a 3.42\% accuracy higher than the compared method. While in the last session, it solely outperforms by 3.83\%. This means that the performance retain is only slightly better. In other words, the proposed approach benefit from a stronger baseline (the first session).

**Summary Of The Paper:**

This study involves a few-shot class incremental learning method, inspired by neural collapse inspired feature-classifier alignment. By using fixed prototype vectors to replaced the learned vectors, which have no constraints, it reduces the confusion and collapse caused by incremental classes with few-shot training samples. Experiments on mini-ImageNet and CIFAR-100 validate the effectiveness of the proposed approach. Overall, this is a piece of interesting work, considering an important yet unsolved problem in the machine learning and computer vision areas.

**Summary Of The Review:**

 While I approve the novel idea of this submission, I am not convinced by the formulation of neural collapse. The performance retain across training session is an important metric, which is missed in the experiments.

---

> ### Author Response · Authors · 2022-11-16
> **Response to Reviewer mQxd (1)**
>
> Thank you for recognizing the contributions of our paper and your valuable feedback.
>
> > (1) Clarify neural collapse and the fixed feature space.
>
> We guess that your comment “the neural collapse illustrated in Fig 1 is built on the assumption of a fixed feature space” is derived from the features being fixed on the same location of a hypersphere across sessions in Fig 1. We would like to clarify this and explain neural collapse in more details here.
>
> In FSCIL, the training data in each novel session is few-shot. Finetuning the backbone network on the few-shot data will cause severe over-fitting on novel classes and forgetting of old classes. As a result, most studies in FSCIL, choose to fix the backbone network after the base session training. So, the features of old classes output by the backbone network are fixed in the later sessions. **That is why Fig. 1 looks like the feature space is fixed. But actually, neural collapse does not require any assumption of fixing the feature space.** We use Fig. 1 to show our motivation. In prior studies, the prototypes (classifier vectors to infer test data) for novel classes are learned in each incremental session, but they may lie close to the old-class prototypes, which causes a small margin for discrimination. To this end, these studies rely on regularizations to shift the old-class prototypes away from the newly learned ones. But this will cause the misalignment between the shifted prototypes and the fixed backbone features for old classes, as shown in Fig. 1 (a). In contrast, our method uses a set of fixed classifier prototypes for all classes in the task. Their structure is instructed by neural collapse to ensure an equiangular separation among classes. In each incremental session, we finetune the projection layer to drive the backbone features to be aligned with the pre-assigned classifier prototypes, which enjoys a consistent target throughout the incremental training and avoids the misalignment, as shown in Fig. 1 (b).
>
> Neural collapse indicates that the feature means and classifier prototypes will be aligned and formed as a simplex ETF, after training a model on a general classification problem of $K$ classes with a $d$-dimension feature space.
> **It is only about the geometric structure of feature and classifier in a well-trained model, and has no relation to the confusion performance of the classifier prediction. Note that the only requirement is that $d \ge K-1$, and the feature space $\mathbb{R}^d$ is NOT required to be fixed.**
>
> The simplex ETF structure instructed by neural collapse has the following important properties:
>
> 1) Equiangular: any pair of classifier prototypes has the same cosine similarity of $-\frac{1}{K-1}$, see Eq. (2) in our paper.
>
> 2) Maximal separation: the cosine similarity of $-\frac{1}{K-1}$ corresponds to the largest possible equiangular separation of $K$ vectors in dimension $d$.
>
> We see that neural collapse corresponds to the optimal state for a classification problem because the inner-class variance is collapsed and the inter-class separation is maximized. In this study, we pre-assign such an optimal structure for FSCIL to drive the backbone features into a consistent target throughout the incremental training and avoid misalignment.
>
> We hope that our response could relieve your concerns. Please let us know if we misunderstand any part of your questions or anything is still confusing.

---

> > ### Author Response · Authors · 2022-11-16
> > **Response to Reviewer mQxd (2)**
> >
> > > (2) Performance retain is only slightly better. The method benefits from a stronger baseline.
> >
> > Admittedly, our performance retain is not so advantageous over ALICE. But we think that the more important metric in FSCIL should be the last-session performance, because the inference in the last session tests the average performance of all classes and has the largest number of old classes. So, the final session performance has the most aggressive demand for not forgetting old classes. Actually, getting a stronger performance on the base session is not a difficult thing, e.g., by using a large backbone architecture or tunning the training setting. But the core challenge in FSCIL is how to keep the good performance of the base session into the last session. That is why the base session performances are not the same across current studies, and researchers usually do not use large backbone architectures in FSCIL. Blindly improving the base session performance does not necessarily lead to a satisfactory result in the last session, and in some cases, it even causes a worse final result because the network is more fitted on the base session classes. For an example, on CIFAR-100, MetaFSCIL has a better base session performance than LIMIT, but is worse than LIMIT in the last session.
> > So, we hold that performance retain should not be the only metric to judge a method. Besides, ALICE (ECCV’22) is a recent strong baseline. On miniImageNet, we have an improvement of 3.42% on the base session and an average improvement of 3.83% across all sessions, which means that the advantage of our method indeed gets larger during the incremental training.
> >
> > > (3) Code
> >
> > We will release our project repo in the final paper, including the code, the trained models, the training logs, and the docker file for re-producibility.

---

> > ### Comment · Reviewer_mQxd · 2022-12-07
> > **I keep the positive score unchanged.**
> >
> > The authors clearly responded to all my review comments. In specific, they clarified neural collapse and the fixed feature space and gave more detailed expain about the simplex ETF structure. I keep the positive score unchanged.

---

### Official Review · Reviewer_sCMo · 2022-10-25

**Confidence:** 4
**Clarity, Quality, Novelty And Reproducibility:** This paper is reproducible and the id…
**Correctness:** 3
**Technical Novelty And Significance:** 2
**Empirical Novelty And Significance:** 2
**Recommendation:** 6

**Strength And Weaknesses:**

Strength:
(1) I think the paper is well organized, clearly motivated, and contributes novelty.
(2) Using Neural Collapse for FSCIL is inspiring, and the proposed loss function to drive the output features to the corresponding prototypes is novel.

Weaknesses:
(1) Unfair comparison. From Table.5 and Appendix: IMPLEMENTATION DETAILS, this method, and competitors use different network architectures. And even this method use model weights pre-trained on ImageNet on the CUB benchmark. However, all other methods (FACT, LIMIT, etc.) train the model from scratch.
(2) This method uses memory to save old class prototype features. However, other methods (FACT, LIMIT, etc.) do not save any data. What's more, the feature extractor is fixed after the base phase, saving features is the same as saving raw data.


**Summary Of The Paper:**

The setting of this paper is few-shot class incremental learning (FSCIL). The key idea is using neural collapse to do FSCIL. This paper use a set of pre-defined classifier prototypes as ETF for the whole label space, and then aligns all classes, including base session's classes and novel classes, to the pre-defined classifier prototypes. What's more, a novel loss function is proposed to drive the features into the corresponding prototypes. Experiments show that the proposed method achieves state-of-the-art performances on three standard benchmarks.


**Summary Of The Review:**

This paper uses neural collapse for FSCIL and obtain good results. Nonetheless, there is a concern about unfair comparison.

---

> ### Author Response · Authors · 2022-11-16
> **Response to Reviewer sCMo (1)**
>
> Thank you for recognizing the contributions of our paper and your valuable feedback.
>
> > (1) Unfair comparison. The method and competitors use different network architectures. The method uses pretrained model on ImageNet for the CUB experiment. The reviewer believes that other methods (FACT and LIMIT) train from scratch.
>
> - About network architecture
>
> When we conducted our experiments, we noticed that backbone architectures on miniImageNet and CIFAR-100 are not consistent in prior studies, as shown in Table 5 in our paper. So, we chose the shallowest one (with the least layers following C-FSCIL [1]), i.e., ResNet-12, for miniImageNet and CIFAR-100. Network architectures for CUB-200 are exactly the same across methods, i.e., ResNet-18 **pretrained on ImageNet**.
>
> In order to relieve your concern, we also use ResNet-18 as the backbone architecture for miniImageNet and CIFAR-100. The results are shown as follows.
>
> On miniImageNet
> | Methods | 0 | 1 | 2 | 3 | 4 | 5 | 6 | 7 | 8 |
> | --- | --- | --- | --- | --- | --- | --- | --- | --- | --- |
> | C-FSCIL (R-12) | 76.40 | 71.14 | 66.46 | 63.29 | 60.42 | 57.46 | 54.78 | 53.11 | 51.41 |
> | ALICE (R-18) | 80.60 | 70.60 | 67.40 | 64.50 | 62.50 | 60.00 | 57.80 | 56.80 | 55.70 |
> | NC-FSCIL (R-12) | 84.02 | 76.80 | 72.00 | 67.83 | 66.35 | 64.04 | 61.46 | 59.54 | 58.31 |
> | NC-FSCIL (R-18) | 84.73 | 78.52 | 74.34 | 70.47 | 68.60 | 65.07 | 61.81 | 60.03 | 58.65 |
>
> On CIFAR-100
> | Methods | 0 | 1 | 2 | 3 | 4 | 5 | 6 | 7 | 8 |
> | --- | --- | --- | --- | --- | --- | --- | --- | --- | --- |
> | C-FSCIL (R-12) | 77.47 | 72.40 | 67.47 | 63.25 | 59.84 | 56.95 | 54.42 | 52.47 | 50.47 |
> | ALICE (R-18) | 79.00 | 70.50 | 67.10 | 63.40 | 61.20 | 59.20 | 58.10 | 56.30 | 54.10 |
> | NC-FSCIL (R-12) | 82.52 | 76.82 | 73.34 | 69.68 | 66.19 | 62.85 | 60.96 | 59.02 | 56.11|
> | NC-FSCIL (R-18) | 82.90 | 77.32 | 73.77 | 69.37 | 65.39 | 62.12 | 59.84 | 57.64 | 55.53 |
>
> It is shown that the results of our method with ResNet-12 and ResNet-18 have similar performances. On miniImageNet, our method using ResNet-18 is better than ResNet-12. On CIFAR-100, although our method with ResNet-18 gets slightly worse in the final performance, it is still the SOTA performance and better than C-FSCIL [1] and ALICE [2].
>
> - About pretraining on ImageNet for CUB
>
> We have to rectify your misunderstanding that “FACT [3] and LIMIT [4] do not use pretrained model and they train from scratch”. Actually, using ImageNet pretrain for experiments on CUB-200 has been widely adopted for not only FSCIL, but also other CV tasks, such as fine-grained recognition.
>
> Although some studies did not explicitly mention this in their paper, nearly all studies in FSCIL followed the CUB-200 training setting of TOPIC [5] (Tao et al.), which is the first FSCIL paper and uses ImageNet pretrained ResNet-18 for CUB-200. Please see their comment in their official code:
> https://github.com/xyutao/fscil/issues/11#issuecomment-687548790
>
> Besides, please have a look at the lines in the official code of FACT [3]:
> https://github.com/zhoudw-zdw/CVPR22-Fact/blob/60262a48a69d0b5ace5693309841896d6226ce21/models/fact/Network.py#L24
> and LIMIT [4]:
> https://github.com/zhoudw-zdw/TPAMI-Limit/blob/c21227f17836fa92db3c7b6ee147e3d605d220db/models/base/Network.py#L25.
> Both the two lines pass ``Ture`` for the ``pretrain`` argument in the ``ResNet`` function. This is a direct proof that they indeed use ImageNet pretrained model for CUB-200 experiments.
>
> As a conclusion, **all methods (as far as we know) in FSCIL use ImageNet pretrain for experiments on CUB-200**. We follow the exact implementation settings as C-FSCIL [1] including the choice of backbone architecture (ResNet-12) for fair comparison. We also use ResNet-18 for experiments on miniImageNet and CIFAR-100 in this response, and the results show that the performances are similar to ResNet-12 reported in our paper.
> We hope our response could eliminate your concern about the fairness of our experiments.

---

> > ### Author Response · Authors · 2022-11-16
> > **Response to Reviewer sCMo (2)**
> >
> > > (2) Our method uses a memory to save old class prototype features. The reviewer believes that other methods (FACT and LIMIT) do not save any data.
> >
> >
> > Some studies indeed do not save prototype feature. **But actually, most of the studies in FSCIL, including FACT [3] and LIMIT [4], save the prototype feature for training or inference**. We summarize as follows:
> >
> > - Save prototype for training: NC-FSCIL (Ours), TOPIC, IDLVQ, Regularizer, C-FSCIL, Sematic-aware KD;
> >
> > - Save prototype for inference: LIMIT, FACT, ALICE;
> >
> > - No memory of prototype: Self-promoted, CEC, MetaFSCIL, Data-Free;
> >
> > Our framework follows C-FSCIL that saves one prototype feature (within-class mean) of each old class to finetune the projection layer. TOPIC, IDLVQ, Regularizer, Semantic-aware KD also use a memory of prototype feature and involve them in training by loss functions or regularizations.
> > The other stream of studies (LIMIT, FACT, ALICE) follows the practice of ProtoNet [6], which is widely adopted in few-shot learning. They use the prototype feature of each class as the classifier vector to infer test data. Please see Sec 4.2 in FACT and Eq. (5) in LIMIT.
> >
> > Therefore, both the two streams (use prototype feature for training or inference) need to save the prototype feature after each session’s training. There is no essential difference between the two streams because the content to store and the memory requirement are exactly the same, i.e., one feature vector for each class.
> >
> > As suggested by Reviewer NNii, we have marked the methods that use a memory of prototype feature in Table 1 and 2 in our revised paper.
> >
> >
> >
> >
> > ### References:
> >
> > [1] Hersche et al., Constrained few-shot class-incremental learning, CVPR 2022.
> >
> > [2] Peng et al., Few-shot class-incremental learning from an open-set perspective, ECCV 2022.
> >
> > [3] Zhou et al., Forward compatible few-shot class-incremental learning, CVPR 2022.
> >
> > [4] Zhou et al., Few-shot classincremental learning by sampling multi-phase tasks, TPAMI, 2022.
> >
> > [5] Tao et al.,  Few-shot class-incremental learning, CVPR 2020.
> >
> > [6] Snell et al., Prototypical Networks for Few-shot Learning, NeurIPS 2017.

---

### Author Response · Authors · 2022-11-30
**General response**

Dear all reviewers:

Thanks again for the efforts in reviewing our paper and the valuable comments. We are encouraged that all reviewers found our work interesting, novel, or inspiring. We have provided a detailed response to each reviewer. Please let us know if there is more question or any concern is still remaining.

Best regards,

Authors

---

### Comment · Area_Chair_AX1v · 2022-12-06
**Question about experiment**

Dear authors,

in your response to NNii you repeat an experiment where you set the number of class prototypes to 500, while in reality there exist only 100 prototypes/classes, and show the results are on par. While this is a promising result, doesn't it conflict with the whole idea of neural collapse being relevant to few-shot classification incremental learning? What stops you from having 1,000 or 10,000 prototypes, and in that case what is the contribution of a neural collapse framework? Won't the cosine distances between prototypes vary (to multiples of 1/K, K being the number of prototypes), effectively returning to a standard non-neural collapse setting?

Cheers,
Your AC

---

> ### Author Response · Authors · 2022-12-06
> **Response to AC's Question**
>
> Dear AC,
>
> Thanks for your efforts in judging our paper and the interesting question.
>
> > Q1&2: Does it conflict with the whole idea of neural collapse being relevant to FSCIL? What stops the authors from having 1,000 or 10,000 prototypes, and in that case what is the contribution of a neural collapse framework?
>
> The inspiration from neural collapse for FSCIL does not rely on the number of prototypes. Neural collapse indicates the following two properties for an ideal classifier:
>
> 1) The prototypes have equi-angular separation, i.e., any pair of two different prototypes has the same cosine distance;
>
> 2) The pair-wise cosine distance is $-\frac{1}{K-1}$.
>
> Although $-\frac{1}{K-1}$ corresponds to the largest possible equi-angular separation of $K$ vectors in a high-dimension space, the most important thing (especially for FSCIL) is the first property. As described in our paper, learning incrementally with a learnable classifier in existing studies suffers from the misalignment between the feature and classifier of old classes. We quote the corresponding explanations here.
> > At each new session, the newly-learned prototypes for novel classes may lie close to the old-class ones. To this end, many studies introduce regularizers to shift the old-class prototypes to make them separated from the novel-class ones with enough distance, e.g. Eq. (7) in C-FSCIL [2]. As a result, there will be a misalignment between the shifted prototypes and the fixed backbone features for old classes. That is why many studies will use another regularizer to prevent the old-class prototypes from shifting far away from their original position, e.g., Eq. (9) in C-FSCIL [2].
>
> But no matter how sophisticated the regularizers are, they cannot ensure an equiangular separation for the whole label space. **As a comparison, we pre-assign a neural collapse inspired structure (ETF classifier) for the whole label space, and train the model towards the same target throughout the incremental training to avoid conflict among sessions.** So, we do not rely on any regularizer. **This is the most important contribution of our neural collapse inspired framework, which is regardless of the number of prototypes.**
>
> Usually, we know the total classes $K$ of a dataset in FSCIL. In this case, we use $K$ prototypes for the ETF classifier because $-\frac{1}{K-1}$ corresponds to the largest possible equi-angular separation of $K$ vectors in a high-dimension space. But when we do not know the total classes, as posed by Reviewer NNii, we can use a large number of prototypes, $K_M$. In this case, **the first property (equi-angular separation) is still satisfied.** The only difference is that the pair-wise cosine similarity is increased from $-\frac{1}{K-1}$ to $-\frac{1}{K_M-1}$. But the two values are very close. They are both near 0 and the structures are both near to be orthogonal. So, as long as the number of prototypes in our ETF classifier is no less than the true class count, it does not adversely affect the performance.
>
>
> > Q3: Won’t the cosine distances between prototypes vary (to multiples of 1/K, K being the number of prototypes), effectively returning to a standard non-neural collapse setting?
>
> Note that even when we use $K_M$ (1,000 or 10,000) prototypes, the $K_M$ prototypes still satisfy neural collapse, i.e., any pair of prototypes has the same cosine similarity. So, the first property is still satisfied and it is still the neural collapse structure. As responded to the last question, the most important contribution is to pre-assign such a neural collapse structure, and drive the backbone features towards the same target throughout the incremental training.
>
> **As a conclusion, using 1,000 or 10,000 prototypes does not return to the non-neural collapse setting. It is still a neural collapse structure, and accordingly, it does not conflict with our neural collapse inspired framework.**
>
> Please let us know if any part still confuses you.
>
> Best,
>
> Authors

---

### Decision · Program_Chairs · 2023-01-20

**Decision:**

Accept: notable-top-25%

**Justification For Why Not Higher Score:**

Combination of two fields, on how an interesting theoretical observation applies to a practical learning setting, is a reason why the paper should be in the spotlight.

**Justification For Why Not Lower Score:**

The core theoretical idea is not new.

**Metareview: Summary, Strengths And Weaknesses:**

The paper investigates the recent observation for neural collapse and how it can be combined with few-shot class incremental learning. The idea is to align a set of prototypes during neural collapse with prototypes required for few-shot learning, and like that benefits experimentally. The idea of neural collapse is obviously not new, however, the application of it to few-shot learning is new, and results are strong.

**Note From Pc:**

if the above contains the word "oral" or "spotlight" please see: "oral" presentation means -> notable-top-5% and "spotlight" means -> notable-top-25%. As stated in our emails, we are disassociating presentation type from AC recommendations